# The Therapeutic Potential of Phytochemicals Unlocks New Avenues in the Management of Rheumatoid Arthritis

**DOI:** 10.3390/ijms26146813

**Published:** 2025-07-16

**Authors:** Kalina A. Nikolova-Ganeva, Nikolina M. Mihaylova, Lidiya A. Kechidzhieva, Kristina I. Ivanova, Alexander S. Zarkov, Daniel L. Parzhanov, Momchil M. Ivanov, Andrey S. Marchev

**Affiliations:** 1Department of Immunology, The Stephan Angeloff Institute of Microbiology, Bulgarian Academy of Sciences, 1113 Sofia, Bulgaria; nikolova_k@microbio.bas.bg (K.A.N.-G.); mihaylova_n@microbio.bas.bg (N.M.M.); l.kechidzhieva@microbio.bas.bg (L.A.K.); 2Laboratory of Eukaryotic Cell Biology, Department of Biotechnology, The Stephan Angeloff Institute of Microbiology, Bulgarian Academy of Sciences, 139 Ruski Blvd., 4000 Plovdiv, Bulgaria; k_ivanova@microbio.bas.bg; 3Faculty of Chemistry and Pharmacy, Sofia University, 1 James Bourchier Ave., 1164 Sofia, Bulgaria; azarkov@uni-sofia.bg; 4Department of Biotechnology, University of Food Technologies Plovdiv, 26 Maritsa Blvd., 4002 Plovdiv, Bulgaria; danielqo_p@abv.bg (D.L.P.); momchox77@gmail.com (M.M.I.)

**Keywords:** rheumatoid arthritis, risk factors, pathogenesis, signaling pathways, medicinal plants, natural products, in vitro and in vivo models

## Abstract

Rheumatoid arthritis (RA) is a progressive and systemic autoimmune disease, characterized by a chronic inflammatory process, affecting the lining of the synovial joints, many body organs/systems, and blood vessels. Its pathological hallmarks are hyperplasic synovium, bone erosion, and progressive joint destruction. Rheumatoid arthritis affects over 20 million people, with a worldwide prevalence of 0.5–1.0%, exhibiting gender, ethnic, and geographical differences. The progressive disability severely impairs physical motion and quality of life and is finally leading to a shortened life span. The pathogenesis of RA is a complex and still poorly understood process in which genetic and environmental factors are principally associated. Current treatment mostly relies on conventional/non-biological disease-modifying anti-rheumatic drugs (cDMARDs), analgesics, non-steroidal anti-inflammatory drugs, glucocorticoids, steroids, immunosuppresants, and biologic DMARDs, which only control inflammation and pain. Along with side effects (drug toxicity and intolerance), these anti-rheumatic drugs possess limited efficacy. Therefore, the discovery of novel multi-target therapeutics with an improved safety profile that function as inhibitors of RA-linked signaling systems are in high demand, and this is in the interest of both patients and clinicians. Plant-derived extracts, nutritional supplements, dietary medicine, and molecules with anti-inflammatory activity represent promising adjuvant agents or alternatives for RA therapeutics. This review not only aims to discuss the basic features of RA pathogenesis, risk factors, and signaling pathways but also highlights the research progress in pre-clinical RA in in vitro and in vivo models, revealing new avenues in the management of the disease in terms of comprehensive multidisciplinary strategies originating from medicinal plants and plant-derived molecules.

## 1. Introduction

Rheumatoid arthritis (RA) is the most common progressive and systemic autoimmune disease, characterized by a chronic inflammatory process, which predominantly affects the lining of the synovial joints (articular damage), a variety of other body organs/systems, and blood vessels (extra-articular manifestations) [1,2,3]. The pathological hallmarks of RA are defined by hyperplasic synovium, bone erosion, and progressive joint destruction due to autoantibody production towards immunoglobulin G (IgG, named rheumatoid factor (RF)) and citrullinated proteins (anti-citrullinated protein antibodies (ACPAs)), carbamylated proteins (anti-carbamylated protein antibodies (anti-CarP)), and lately, acetylated proteins (anti-acetylated protein antibodies) [4,5,6]. Both RF and ACPAs have approximately equal sensitivity and specificity to RA. The disease is also marked by elevated levels of C-creative protein (CRP) and a high erythrocyte sedimentation rate (ESR) [7]. The most common serum prognostic biomarkers for RA are RF and ACPA since their appearance is observed approximately 4.5 years prior to the clinical onset of the disease. A relatively new potential biomarker for RA, revealing high specificity, is the oncoprotein survivin, which is detected in 50.7% of RA patients and only in 5.6% of controls [8]. Nevertheless, 50–80% of RA patients harbor autoantibodies and are seropositive, while other patients are seronegative for these autoantibodies. For that reason, to identify RA patients, clinical diagnostics, e.g., physical examination, clinical symptoms, antibodies present in the blood, and imaging findings, needs to be applied [9,10,11].

### 1.1. Epidemiological Overview and Global Prevalence

Although RA can occur at any time, it is mainly manifested in elderly populations, peaking between the ages of 40 and 70 [12]. Rheumatoid arthritis affects over 20 million people, and its average worldwide prevalence is about 0.5–1.0%, exhibiting gender (the ratio of female to male prevalence is 3:1), ethnic, and geographical differences, therefore significantly varying among different populations [7,12]. Along with joint damage, there are other extra-articular co-morbidities affecting the cardiovascular, endocrine, neurological, ocular, and pulmonary systems, including hematologic, renal, and hepatic disorders [13]. The progressive disability severely impairs physical motion and quality of life, finally leading to a shortened life span [6], reporting mortality rates twice higher in patients with RA, and this number is increasing [14]. Based on analyses for the period of 1990–2020, it was reported that in 2020, there were 17.6 million RA patients worldwide, and the age-standardized global prevalence rate increased by 14.1% since 1990. The age-standardized death rate decreased by 23.8% from 1990 to 2020, while the disability-adjusted life years (DALYs) increased by 76.4%. In spite of that, RA is still a key public health concern worldwide. The global burden of RA has increased in recent decades and will continue to increase, and it is forecasted that 31.7 million individuals will be living with RA by 2050 [15]. According to epidemiological forecasts, the total number of cases of RA will continue to increase, with up to a 1.36% annual growth rate in 2025. The US is expected to have the highest number of diagnosed cases, followed by Japan and Australia [16]. The global RA market recorded a historic compound annual growth rate (CAGR) of 4.9% in the past 6 years from 2019 to 2024 [17], and it is expected that the top seven major RA markets (US, EU4, UK, and Japan) will secure a CAGR of 2.17% in the coming years from 2025 to 2032 (from USD 28.0 billion to USD 35.4 billion). Among all DMARDs, JAK inhibitors and monoclonal antibodies are driving market expansion [18].

Rheumatoid arthritis has a negative impact on the most productive and active years (30–50 years) of an individual’s lifespan [19], severely affecting the quality of daily life, including high disability rates and potential loss of labor [20]. Socioeconomic surveys indicated that people with severe (60%) or moderate (48%) pain experience additional work obstacles compared with dose with mild (34%) or no (19%) pain, and a significant correlation was found between severity, pain, disability, and early retirement [21]. The major economic costs for RA patients are based on both healthcare expenditures and loss of productivity [22]. For example, the average annual cost of medical care for RA depending on the medication may vary between USD 12,500 and USD 36,000, while the Rheumatoid Arthritis Support Network estimates that low productivity, absenteeism, and lost wages can cost from USD 1500 to USD 22,000 per year per patient [23]. In addition to the joints, many advanced patients develop systemic and comorbid extra-joint diseases, such as hypertension, diabetes mellitus, ischemic heart disease, chronic obstructive pulmonary disease, asthma, tuberculosis, chronic liver disease, hypothyroidism, hyperthyroidism, active malignancy, etc., which significantly reduce life expectancy [7,24].

The clinical manifestation of RA differs in its early and untreated/inadequately treated late stage. Typical symptoms of the early stage are fatigue, swollen and tender joints, redness, and stiffness not only in the morning but also after a period of inactivity. On the other hand, the late stage is characterized by more severe manifestations, such as swan neck deformity, ulnar deviation, and subcutaneous nodule formation [6,25], as well as bone corrosion, muscle atrophy, synovitis invasion of articular cartilage, sub-cartilage bone erosion, and damage to the ligaments and tendons [20], which may finally result in premature death [15]. Rheumatoid arthritis has a multi-synovial form whose primary clinical manifestation is repeated and symmetrical multiple micro arthritis [20]. It is a form of polyarthritis due to its involvement of multiple joints (typically six or more), mainly affecting the hands, wrists, knees, feet, and metacarpophalangeal, proximal interphalangeal, and metatarsophalangeal joints. Erosions of the feet and hands are observed radiographically [20,26]. The pathogenesis of RA is a complex and still poorly understood process in which genetic and environmental factors are principally associated with the disease onset and progression [24,26]. Joint inflammation is triggered and maintained by invasion and interaction between several types of immune cells, including neutrophils, antigen-presenting cells (B cells, macrophages, and dendritic cells), T cells, and fibroblast-like synoviocytes (FLSs) [19,27]. These chain events contribute to the release of pro-inflammatory cytokines (such as interleukin-1 (IL-1), -6, and -17; prostaglandins; and tumor necrosis factor alpha (TNF-α)), which are mediators that activate additional signaling pathways, including the upregulation of cyclooxygenase-2 (COX-2), matrix metalloproteinases (MMPs), Janus kinase signal transducers and activators of transcription (induced by IL-6), and the nuclear factor kappa-B (NF-κβ) pathway (induced by IL-1 and TNF-α), which continue to generate and accumulate inflammation [19,27]. The above-mentioned cells and cytokines are involved in the RA microenvironment (consisting mainly of the extracellular matrix (ECM) and stromal cells), where the generated inflammation creates a hypoxic environment and subsequently triggers the generation of reactive oxygen species (ROS) and angiogenesis, and as a consequence, the formed microvessels cause the synovial membrane to invade the cartilage surface and to form pannus, destroying the structure and function of bone and cartilage [28].

Rheumatoid arthritis is often regarded as incurable, and currently, no adequate treatment is available [6]. Current treatments aim to control inflammation and pain, contributing to patients’ symptom relief and therefore avoiding or minimizing joint destructive processes [7]. The current treatment of RA mostly relies on conventional/non-biological disease-modifying anti-rheumatic drugs (cDMARDs), analgesics, non-steroidal anti-inflammatory drugs (NSAIDs), glucocorticoids (GCs), steroids, immunosuppressants, and biologic DMARDs (bDMARDs) [7,12,29]. However, in most cases, the drugs need to be accepted weeks or months before they have any effect, and they may cause liver and bone marrow toxicity, certain malignance, and/or have opportunistic effects [12,30]. Patients are considered as “difficult to treat” or “refractory” to RA, and although targeted synthetic DMARDs have been introduced, RA treatment is still a great therapeutic challenge, and a considerable amount of patients remain symptomatic [31]. Along with side effects, such as drug toxicity and intolerance, these anti-rheumatic drugs possess limited efficacy; therefore, novel multi-target therapeutics with an improved safety profile are in high demand, and this is in the interest of both patients and clinicians. The use of plant-derived remedies offers several potential advantages such as anti-inflammatory and immunomodulatory properties, potential pain relief, and the possibility of reducing reliance on conventional medications. However, several factors regarding its effectiveness and safety should be regarded, particularly possible interactions with conventional medications. Other considerations include challenges with bioavailability, metabolism, and the standardization of herbal remedies [12]. Plant-derived extracts, nutritional supplements, dietary medicine, and molecules with anti-inflammatory activity represent promising adjuvant agents or alternatives for RA therapeutics [12]. The pathogenesis of RA is tightly related to many characterized signaling pathways, and research attention has focused on discovering plant-derived molecules that function as inhibitors of RA-linked signaling systems [20].

This review first attempts to recapitulate the dynamics of RA pathogenesis, highlighting the role of risk factors, effector cells involved, the role of metalloproteinases, oxidative and nitrosative stress, angiogenesis, cell migration and invasion, and the production of cytokines and chemokines. Next, we discuss existing conventional therapies for RA and their limitations. A summary of various signaling pathways involved in RA development is performed, outlining the efficacy and mechanism of plants and their phytoconstituents in RA management, supported by preclinical data based on state-of-the-art in vitro and in vivo models. Finally, we analyze the existing theoretical and practical challenges and unanswered questions in the field, aiming to improve RA therapy using natural products.

### 1.2. Risk Factors

Unlike osteoarthritis, RA does not develop due to age, but its onset and development are rather a result of a combination between immune dysfunction and hereditary (genetic factors) and environmental factors. Because of its complexity, it has not been thoroughly clarified yet; therefore, research into the genes, pathways, and pathogenic immune cell subsets in RA has advanced the understanding of the mechanisms involved in pathogenesis [19,30]. The major differences between healthy and RA joints, as well as its process of establishment, are illustrated in Figure 1.

### 1.3. Genetic Factors

The contribution of genetic predisposition is thought to be about 50 to 60%, which has the most significant impact on the vulnerability to RA [32]. Genetic studies have been successful in determining the heritable component of RA susceptibility and outcome, of which RA severity and response to treatment are important [33]. The presence or absence of RF and ACPAs classifies RA into seropositive and seronegative, and there are differences between the risk factors involved. Tyrosine phosphatase non-receptor type 22 (PTPN22) risk alleles [34,35], human leukocyte antigen D-related (HLA-DR) alleles [36], and tumor necrosis factor-receptor associated factor 1 (TRAF1) and complement component 5 (TRAF1/C5)-related genes are the main genetic factors associated with an ACPA-positive subtype [37], while interferon regulatory factor 5 (IRF-5) is confined to be associated with the ACPA-negative subtype [38]. Despite the lack of concrete evidence supporting a direct role in RA pathology, some genetic markers regulate immune responses and account for the variations in RA susceptibility [24]. The strongest genetic association with RA susceptibility is located at the HLA locus. This is a specific HLA class II gene, also known as a major histocompatibility complex (MHC) loci, encoding MHC molecules that may contain the shared epitope [4]. Individuals carrying a single HLA allele have a five-fold higher risk of developing RA than individuals without this gene [39]. The susceptibility and outcome of RA may be related to specific HLA-DR alleles; however, these alleles vary by ethnicity and geographic region [40,41]. The HLA-DRB1 gene constitutes the strongest genetic association linked to RA, and the allele associated with the disease is named as a shared epitope, which is a conserved sequence of five amino acids at positions 70–74 of the HLA-DRB1 gene [20], and this concept has been highly correlated with the ACPA-positive RA [33]. The shared epitope hypothesis indicates that some alleles with this conserved sequence are linked with the pathogenesis of RA since they allow antigen-presenting cells to incorrectly present their antigens to T cells, which results in T cell-mediated autoimmune responses that directly contribute to the RA pathogenesis [42].

On the other hand, single nucleotide polymorphisms (SNPs) are variations in the DNA code at a specific position in the genome, and these variations have been associated with susceptibility to diseases, including RA [33]. SNPs have been detected in T cells regulating the HLA-gene. In the presence of SNPs, T cells induce tissue damage and inflammation linked to RA during the rapid resolution of joint defects, and therefore, its detection may enable an assessment of the risk of developing RA and hereditary autoimmune disorders in general [43]. The specific RA-related gene HLA-DR4 is found in 60% to 70% of patients diagnosed with RA, while it is detected in only 20% of the general population [44]. The HLA-DRB1 gene has been associated with the susceptibility of this disease, especially with shared epitope-coding alleles (HLA-DRB1*0401, *0404, *0405, *0408, *0101, *0102, *1402, and *1001) [45]. In addition, genetic differences between ACPA-positive and ACPA-negative RA have been demonstrated, e.g., variants in *HLA-DRB1*, *PTPN22*, *BLK*, *Ankyrin Repeat Domain 55 (ANKRD55)*, and *IL6ST* are associated with RA regardless of serological status, whereas *AFF3*, *CD28*, and *TNFAIP3* are found only in seropositive RA, and *PRL* and *NFIA* are found only in seronegative RA [46,47]. All of these variants demonstrate very well the RA susceptibility in both positive and negative types of RA. However, it is equally important to identify markers of disease severity. In this regard, it has been demonstrated that several markers for susceptibility are also associated with severity, e.g., *HLA-DRB1*, *IL2RA*, *DKK1*, *GRZB*, *MMP9*, and *SPAG16* [46,48], although some of them, such as *FOXO3*, are associated with severity alone [46,49]. There are also other HLA loci independently associated with RA susceptibility: amino acid position 9 of HLA-DPB1 (another HLA class II gene) [50] and two associations within HLA class I genes, such as amino acid position 9 of HLA-B [50] and position 77 of HLA-A [51]. Along with the HLA gene, there are many genetic variations outside of the HLA complex related to RA, such as *PTPN22*, *STAT4*, *TRAF1-C5*, *CTLA4*, and *PADI4* [52]. The *PTPN22* gene encodes for the cytoplasmic lymphoid specific tyrosine phosphatase (Lyp). This enzyme has gained enormous interest due to a genetic SNP of its gene *PTPN22* rs2476601 (R620W), which has been associated with several human autoimmune diseases, including rheumatoid arthritis (RA). Lyp has been shown to be a powerful inhibitor of T cell and B cell activation by binding to the SH3 domain of the Csk tyrosine kinase, an important negative regulator of T cell and B cell antigen receptor signaling [53,54,55,56]. Along with that, the role of Lyp in TNFα-induced priming of neutrophil ROS production has been investigated during RA development. It has been demonstrated that Lyp-selective inhibitors inhibited TNFα-induced priming of neutrophil superoxide anion production through the inhibition of key pathways involved in neutrophil priming, such as the inhibition of p47phox phosphorylation on Ser345, ERK1/2 phosphorylation, and Pin1 [57].

### 1.4. Epigenetic Factors

Genetic heterogeneity cannot explain all aspects of RA; therefore, an examination of epigenetic factors and mechanisms might be of ultimate importance for the provision of novel therapeutic factors [58]. Epigenetics are heritable changes in gene expression without altering the DNA sequence, but epigenetics determines which genes are turned on and/or off, which mainly include histone modification, DNA methylation, and non-coding RNA mechanisms, and these processes can be affected by different genetic and environmental factors. Epigenetic modifications are reversed processes, and the corresponding enzymes which control histone modification or DNA methylation could be proposed as drug targets for RA [20]. DNA methylation is the most commonly occurring postreplication DNA modification under the activity of DNA methyltransferases (DNMTs), which transfer the methyl group from S-adenosine methionine (SAM) to the DNA sequence. DNA methylation occurs at the cytosine of CpG (cytosine–phosphoric acid–guanine) islands to produce 5MC, most of which are located in the promoter region. Hypermethylation in the promoter region is related to gene silencing or gene inactivation, while its hypomethylation activates transcriptional activity and promotes gene expression of extracellular matrix proteins, growth factors/receptors, matrix-degrading enzymes, and adhesion molecules [59]. For example, several studies show that fibroblast-like synoviocytes (FLS) and peripheral blood mononuclear cells (PBMCs) in RA patients are characterized by extensively hypomethylated genomic DNA. Rheumatoid arthritis fibroblast-like synoviocytes (RA-FLSs) have a unique, non-random methylation pattern-methylome, which is specifically reorganized during disease progression and varies depending on the joint localization [58], and DNA methylation reduction is often found in highly proliferative tissues and is associated with a relative lack of methyl groups’ donor S-adenosylmethionine (SAM) [20]. T-box transcription factor 5 (TBX5) regulates the expression of pro-inflammatory cytokines and chemokines in SF, including CXCL12 (chemokine C-X-C motif ligand 12) chemokine. Upon hypomethylation, *TBX5* increases its own expression and that of *CXCL12*, which is associated with protein accumulation in RA patients and contributes to chronic inflammation [60]. The promoter demethylation of *IL-6* and *IL-10* genes in a single CpG sequence contributes to the increase in cytokine levels as the disease progresses [61]. A comprehensive analysis of DNA methylation in RA-SFs identified 1 859 differently methylated loci, of which hypomethylated ones such as *CHI3L1*, *CASP1*, *STAT3*, *MAP3K5*, *MEFV*, and *WISP3* were of critical importance and found to be involved in cell migration, adhesion, transendothelial penetration, and interactions in the extracellular matrix [62]. It has been reported that the PBMCs from RA patients have a significant overall DNA hypomethylation state compared to healthy people [63]. Analyses of the whole-genome DNA methylation and mRNA expression profiles of PBMCs from patients with RA revealed that approximately 1046 DNA methylation sites were closely associated with the pathogenesis of RA. Among the identified differentially methylated positions (DMPs) and genes, an interferon-inducible gene interaction network (such as *MX1*, *IFI44L*, *DTX3L*, and *PARP9*) was formed. The methylation of PARP9 was correlated with mRNA levels in Jurkat cells and T lymphocytes isolated from patients with RA. The *PARP9* gene exerted significant effects on Jurkat cells (e.g., cell cycle, cell proliferation, cell activation, and expression of inflammatory factor IL-2) [64]. Epigenetic modification in immune cells is also critical for the development of RA. Differentially methylated patterns of B lymphocytes were observed in RA and systemic lupus erythematosus (SLE) patients, and in the control group, CpG sites were located in the *CD1C*, *TNFSF10*, *PARVG*, *NID1*, *DHRS12*, *ITPK1*, *ACSF3*, and *TNFRSF13C* genes [65]. On the other hand, the hypermethylation of 4 CpG dinucleotides in exon 7 of *LRPAP1* has been correlated with patients who demonstrated no response to therapy by TNF inhibitors compared to responders. The *LRPAP1* gene is expressed in PBMCs and encodes the chaperone of low-density lipoprotein receptor-related protein 1, which affects the activity of transforming growth factor beta (TGF-β) [66]. The aberrant function of regulatory T cells (Treg) in RA patients was associated with the hypermethylation of a specific region in the promoter of the cytotoxic T-lymphocyte associated protein 4 (CTLA-4; −658 CpG) in comparison with healthy controls. DNA hypermethylation prevents the binding of the nuclear factor of activated T cells (NF-AT) with the cytoplasmic one, called NF-ATc2, which leads to a decrease in *CTLA-4* expression. As a consequence, Treg cells were unable to induce the expression and activation of the tryptophan-degrading enzyme indoleamine 2,3-dioxygenase (IDO), which in turn resulted in a failure to activate the immunomodulatory kynurenine pathway [67]. Furthermore, treatment with methotrexate induced DNA hypomethylation of FoxP3 locus in Treg. This resulted in gene upregulation with a consequent increase in the CTLA-4 concentration and the normalization of Treg function in RA. These studies clearly illustrate how aberrant DNA methylation can affect cell functions and how epigenetic mechanisms can be used in therapy [68].

Histone modification refers to a post-translational modification of a specific site on histones in chromatin. Acetylation, methylation, phosphorylation, and ubiquitination are all included within the modifications of the histone tails, and among them, acetylation is the most common. Histone deacetylases (HDACs) have a significant role in the activation or silent regulation of pro-inflammatory genes, and their inhibitors are often used to study the pathogenesis of RA. The HDAC activity and histone H3 acetylation status in PBMCs are potential biomarkers for evaluating disease activity [59]. For example, the hyperacetylation state caused by the decreased activity and expression of HDACs promotes pro-inflammatory processes and ultimately leads to RA in PBMCs [69]. Mice with a T cell-specific deficiency of HDAC1 (HDAC1-cKO) were found to be resistant to the development of collagen-induced arthritis (CIA), while its activation produces pro-inflammatory factors. HDAC1 is a promising target for the treatment of RA patients since the inflammatory cytokines IL-17 and IL-6 were significantly reduced in the serum of HDAC1-cKO mice. Along with that, selective HDAC inhibitors restrained chemokine receptor 6 (CCR6) upregulation in a mouse model and human CD4^+^ T cells [70]. Another potential target for RA treatment is SIRT1 (a type 3 histone deacetylase that possesses anti-inflammatory properties), which can reduce the inflammatory responses in RA by regulating M1/M2 macrophage polarization. For example, activated SIRT1 promotes the phosphorylation of adenosine monophosphate-activated protein kinase α (AMPKα)/acetyl-CoA carboxylase in macrophages through the upregulation of M2 genes, such as *MDC*, *FcϵRII*, *MrC1*, and *IL-10* expression. Simultaneously, activated SIRT1 downregulates LPS/γ interferon mediated NF-κB activity by inhibiting p65 acetylation and M1 gene (including *CCL2*, *iNOS*, *IL-12p35*, and *IL-12p40*) expression [71].

MicroRNAs (miRNAs) widely present in all organisms and present endogenous non-coding single-stranded small RNAs of about 22 nucleotides in length [59]. Initially, the miRNA gene is transcribed to form a primary miRNA in the nucleus, cleaved by Drosha to form a precursor miRNA. Exportin-5 transfers miRNA to the cytoplasm, where it is cleaved by Dicer to form mature miRNA duplexes, which are then unfolded, and a miRNA strand is added to the RNA-induced silencing complex and acts as a post-translational repressor of the gene [20]. MicroRNAs have been implicated in the occurrence and progression of many diseases, including RA. For example, hsa-miR-132-3p, hsa-miR-146a-5p, and hsa-miR-155-5p are potential biomarkers of responsiveness to methotrexate (MTX) therapy, whose levels were lower in responders [72]. However, the downregulation of miR-10a in RA-FLs accelerated NF-κB activation and significantly promoted the production of various inflammatory cytokines, including TNF-α, IL-1β, IL-6, IL-8, MCP-1, and matrix metalloproteinase (MMP)-1 and MMP-13 [73]. Some miRNAs have been regulated in RA, and it has been demonstrated that DNA methylation increased the expression of miR-203, which led to the increased secretion of MMP-1 and IL-6 via the NF-κB pathway and contributed to the activated phenotype of RA-FLs [74]. The upregulation of miR-203 induces RA by promoting the generation of MMP-1 and IL-6 [74]. The depletion of miR-19b positively regulates NF-κB signaling through the suppression of a regulon of negative regulators of the same pathway [75].

### 1.5. Environmental Factors

Many environmental factors, including smoking, alcohol, air pollution, and exposure to insecticides, viruses, bacteria, toxic minerals, or mineral oil increase the risk of RA [24]. Most of the factors listed here support the hypothesis that the etiopathogenesis of RA is of “mucosal origin”, in which the autoimmune processes that lead to the development of RA are triggered in the mucosa-associated lymphoid tissues in the lung, the oral cavity, and the gut prior to systemic spread [76]. Rheumatoid arthritis-related autoantibodies may be generated in the lung mucosa and in draining lymph nodes prior to the onset of clinically apparent RA [77,78,79]. Smoking is the strongest environmental factor in RA development and raises the risk of RA in a gradient fashion, with double risk among smokers with a 20-year history of tobacco use compared with nonsmokers. The correlation between tobacco use and RA is the strongest or almost limited to RF- or ACPA-positive individuals [80] with at least one copy of the shared epitope alleles HLA-DR Beta 1 [81] and has no or very little effect on ACPA-negative individuals [82]. The risk of developing RA between individuals with the shared epitope and individuals who smoke can be increased by 20-fold or more compared with nonsmokers who do not carry the shared epitope [83]. Actually, no association between passive smokers and the risk of developing RA has been observed [84]. In addition, the increased risk associated with smoking might be mediated by epigenetic modifications as smoking was significantly associated with the hypomethylation of certain DNA regions [85]. Population-based studies revealed that smoking is a strong risk factor for RA in men rather in women, and this effect might be due to hormonal differences in the modulation of the immune cascade activated by smoking [86].

The impact of air pollution is a topic of increasing concern in general health, and this does not spare rheumatology. In a bidirectional two-sample Mendelian randomization study regarding genetic and causal connections, a significant correlation was found between air pollution, smoking, and the development of RA, where the C-reactive protein (CRP) was identified as a major mediator in the relationship [87]. Another study also confirmed that air pollution, especially the presence of NO_x_, is positively associated with a risk of RA. In risk factor-related mediation analyses, it was found that smoking mediated 9% of the effect of NO_x_ on RA [88]. Long-term air pollution exposure for at least 12 years, especially the presence of particulate matter with a diameter of 25 µm (PM2.5) and NO_2_, was also associated with an increased risk of RA [89]. Even ambient air pollution, especially the presence of SO_2_ and NO_2_, is strongly associated with RA disease activity [90,91].

Anthropogenic pollutants (containing a mixture of CO, NO, NO_2_, particulate matter with a diameter of 10 to 25 µm or less, or O_3_) and their inhalation induce the formation of a particular type of lymphoid tissue known as inducible bronchus-associated lymphoid tissue (iBALT), which is linked to the citrullination of proteins triggering the development of RA [92]. In addition, it was found that the long-term exposure of 10 µg/cm^3^ of NO_2_ (which is equal to four passively smoked cigarettes) increases the risk of RA [93]. In a study including 888 patients with RA, 3396 patient follow-up visits, and 13,636 daily air pollutant visits, an exposure–response relationship was found between the concentrations of air pollutions (CO, NO, NO_2_, NO_x_, PM10, PM2.5, and O_3_) for 60 days, and a higher risk of experiencing a flare of arthritis was observed. The main cause was found to be elevated CRP levels ≥ 5 mg/L [93].

Pulverized cement, silica, asbestos, glass fibers, textile dust, and many other pollutants are all associated with RA development [94,95]. Exposure to silica in different workers (farmers, minors, foundry, construction, granite, coal, and textile workers) is associated with RA development [94,95,96]. For example, there is a correlation between silicosis and RA, which mainly affects patients with ACPA-positive RA and is caused due to silica dust inhalation [97]. Silica-exposed subjects were found to have a moderately increased risk of ACPA-positive RA; however, subjects exposed to rock drilling were found to have a more markedly increased risk of ACPA-positive RA [95]. The association between silica and RA was first described in coal miners who were developing RA and pneumoconiosis, a clinical entity coined Caplan syndrome or rheumatoid pneumoconiosis. A cohort study of the total Danish working population confirmed an increased risk of RA with increasing cumulative exposure to respirable silica particles, and it additionally reporting an increased risk of developing systemic lupus erythematosus and small vessel vasculitis [98,99]. It is still uncertain whether silica-induced inflammation and fibrosis contribute equally to the onset of RA autoimmunity. However, findings in human and animal model studies support an autoimmune process that begins with the activation of pulmonary macrophages, which engulf silica particles in the alveoli, triggering pro- inflammatory cytokine production [98]. In a case–control-study in Malaysia, occupational exposure to textile dust was significantly associated with an increased risk of developing RA in the female population [94]. In a case–control-study it was established that male workers exposed to asbestos had a higher risk of seropositive RA and seronegative RA compared with unexposed workers [100].

Exposure to metals, such as cadmium, is also documented to contribute to RA development, as studied in Korean females. The prevalence of RA in women was increased with increasing quartiles of Cd levels, with a 19-fold difference in female RA prevalence between individuals in the lowest quartile of serum Cd levels and those in the highest quartile [101].

### 1.6. Lifestyle Factors (Nutrition and Gut Microbiota)

It has been suggested that some healthy diets, e.g., the Mediterranean diet, have a modest protective effect in individuals with seropositive RA [102]. The consumption of fruits and fish, including omega-3 fatty acids, has a protective effect related to RA-associated autoimmunity [103,104,105]. Long-term supplementation of omega-3 and vitamin D resulted in a 25–30% lower incidence of RA [106]. Coffee consumption may be a risk factor for RA, with a possible explanation being its involvement in the production of RFs [107]. The intake of raw meat and sugar-sweetened sodas is associated with an increased risk of RA [108,109,110]. Microbiota (periodontal and gut) and infection microorganisms are associated with an increased risk of RA development as well [111]. For example, the oral microbiota *Porphyromonas gingivalis* and *Aggregatibacter actinomycetemcomitans* causing periodontal disease are also related to RA development [112,113]. The dysregulation of many intestinal bacteria is associated with RA onset. For example, the expansion of *Prevotella* species is associated with RA-related autoimmunity and a is marker for early disease preclinical development [114]. Other genera, such as *Bacteroides*, *Eggerthella*, and *Collinsella*, have also been related to RA [115,116,117]. Several infectious agents have been considered as possible causes of RA, including rubella virus, Epstein–Barr virus, and mycoplasma organisms [24].

### 1.7. Personal Factors

Although both males and females are at risk of RA development, RA is more prevalent in women, with a female-to-male sex ratio ranging from 4:1 in younger individuals to less than 2:1 in older populations with the disease [114]. This might be due to hormonal changes (such as from contraceptive use) or a sudden decline in estrogenic function (seen in menopause or with the use of anti-estrogenic therapies) [118,119]. The manifestations of RA improve or even disappear during pregnancy [120]. In women, RA most commonly becomes symptomatic around middle age or at the time of menopause. Men have a later disease onset, are more likely to be positive for RF, and have higher titers of ACPAs [121]. Epidemiologic studies have shown a complex interaction between RA and obesity. Obesity can contribute to the development of seronegative RA, especially in younger women, although it is suggested to be associated with a decreased risk of RA among men [122]. Adipose tissue secretes different pro-inflammatory and anti-inflammatory factors, including the adipokines leptin, adiponectin, resistin, and visfatin, as well as cytokines and chemokines, such as TNF-α and IL-6, which participate in RA development [123].

Sex hormones may impact the development, risk, and course of autoimmune illnesses and several aspects of immune system function. While testosterone and progesterone naturally depress the immune system, estrogens, especially 17-estradiol (E2) and prolactin, operate as enhancers of humoral immunity at least, upregulating the production of immunoglobulins and downregulating inflammatory responses [124]. The inverse correlation between RA severity and androgen levels could be a reason why RA is less severe in men [122]. For example, postmenopausal women had a two-fold increased risk of seronegative RA compared with premenopausal women [118]. Another study concluded that women who experienced menopause before 40 had increased odds of postmenopausal rheumatoid arthritis compared to women experiencing menopause at 50 years of age or later [119].

Female and male immune systems are different as they are mainly affected by the distribution of hormones, the presence of two X chromosomes versus only one, and a singular response to environmental factors. Several genes on the X chromosome regulate innate and adaptive immune responses. In addition, sex hormones are immunoregulatory, participate in the secretion of cytokines and chemokines, interact with inflammatory mediators, and play an essential role in pathobiological differences [125]. Two or more X chromosomes increases the risk for some autoimmune diseases, and the increased expression of some X-linked immune genes is frequently observed in female lymphocytes from autoimmune patients [126]. The X chromosome is an important risk factor for autoimmune susceptibility as 46,XX females have an overall greater risk of developing autoimmunity compared to 46,XY males. Interestingly, women with Turner syndrome (45,X) who have a single X chromosome are underrepresented in cases of female systemic lupus erythematosus (SLE), supporting the hypothesis that multiple X chromosomes increases autoimmune susceptibility [127,128,129]. Men with supernumerary X chromosomes, such as individuals with Klinefelter syndrome (47,XXY), are predisposed toward autoimmunity similar to 46,XX females. The risk of SLE in 47,XXY males is 14-fold higher than 46,XY males with a single X chromosome [130]. Men with Klinefelter syndrome (KS) also have an increased risk of Sjögren’s syndrome (SS), which is also strongly female-biased [131]. As a confirmation, in a cohort study of adults with KS, 47, XXY, non-organ-specific immunoreactivity was found to be significantly higher in patients with 47,XXY KS (14%) than in the controls. Among all the antibodies investigated, only anti-nuclear antibodies (ANAs) were observed significantly more frequently in patients with 47,XXY KS (12.1%) than in the controls [132]. Furthermore, female trisomy (47,XXX) is also overrepresented in SLE, as the prevalence in 47,XXX compared to karyotypically normal 46,XX women is 2.5:1. SS has a similar increased prevalence in females with trisomy compared to normal females (2.9:1) [133].

### 1.8. Mechanisms and (Pato)Etiology of Rheumatoid Arthritis Initiation, Development, and Progression

The pathogenesis of RA is complex and poorly understood in terms of causes, initial triggers, and progression process [19]. Autophagy is a physiological process that functions as a pro-survival mechanism and also protects cells from hazardous dysfunctional molecules, degrade intracellular pathogens, and damaged organelles [134,135,136]. Alterations in the process and regulation of autophagy could be associated with the pathogenesis of various autoimmune diseases, including RA [137]. A number of studies have shown a direct link between autophagy and the generation and presentation of post-translationally modified (PTM) proteins, such as citrullinated, carbamylated, and acetylated proteins. Autophagy facilitates the citrullination process by supporting peptidyl arginine deiminase (PAD) activity in autophagosomes. Antigen-presenting cells like DC and macrophages constitutively exhibit autophagy because of the presentation of citrullinated proteins. Furthermore, the presentation of citrullinated proteins by B cells could be triggered by autophagy induction because of BCR engagement or nutrient deprivation. The accumulation of carbamylated proteins within RA joints leads to the formation of anti-CarP antibodies, which is associated with severe joint pathology [138]. Autophagy enhances the intracellular processing of these proteins [139], promoting their presentation via MHC class II molecules to T cells and amplifying autoimmune responses [140]. Another post-translational modification contributing to RA pathology is acetylation. Altered acetylation leads to abnormal gene expression, inflammation, and the formation of neo-epitopes. The turnover of acetylated proteins is regulated by autophagy, linking dysfunctional protein clearance to disease progression [141,142]. Several studies support the view that post-translational processing of proteins in autophagy may generate autoantigens recognized by the immune system in early active RA. The key role for autophagy in the citrullination of peptides by antigen-presenting cells has been hypothesized. Citrullination in the autophagosomes may increase the catabolism of the proteins, as charged residues of the proteins are eliminated [143,144,145]. Actually, in vivo autophagic cells showed PAD-4 activation with consequent protein citrullination. Ex vivo, a significant association between the levels of autophagy and anti-CCP antibodies was observed in naïve RA patients [146]. Rapamycin induction of autophagy in RA FLS could elevate citrullinated proteins including α-enolase and vimentin. An investigation on monocytes derived from patients in the early stage of RA revealed a direct association between anti-CCP titers and the level of microtubule-associated protein 1A/1B-light chain 3 (LC3-II) [147]. Altogether, these data propose that the presentation of citrullinated proteins could be regulated by the autophagy mechanism. Furthermore, the formation and presentation of citrullinated proteins may lead to the loss of self-tolerance and highlight autophagy as a critical player in the pathogenesis of autoimmune diseases such as RA. In addition, it has been demonstrated that autophagic cells show a significant increase in carbamylated proteins, and a significant correlation was found between autophagy and carbamylation levels in mononuclear cells of naïve RA patients [139].

In RA, dysregulated autophagy contributes to the abnormal survival and aggressive behavior of fibroblast-like synoviocytes (FLSs) and immune cells within the inflamed synovium, promoting synovial hyperplasia and resistance to apoptosis. Compared to healthy FLS, RA-FLSs are characterized by impaired autophagy and residence to apoptosis that is shown by upregulated Beclin1 and LC3II protein levels along with decreased p62 protein levels [148]. Two members of the IL-1 cytokine family, IL-36 and IL-38, are both elevated in RA patients. Both of them have opposite functions according to autophagy. IL-36 accelerates the activation of autophagy and inhibits autophagy-restrained proliferation, migration, and invasion in synovial cells. Conversely, IL-38 inhibits autophagy and promotes proliferation, migration, and invasion [149]. This leads to the excessive accumulation of inflammatory mediators such as IL-2, IL-6, IL-13, and IL-17; local inflammation; and joint destruction.

Bone erosion, a hallmark of RA, is primarily mediated by osteoclasts, which exhibit high levels of autophagy that are correlated with disease severity [150]. Chronic inflammation mediated by cytokines like RANKL, TNF-α, and IL-6 stimulates both autophagy and osteoclastogenesis, leading to bone resorption. In contrast, impaired autophagy in osteoblasts may contribute to reduced bone formation, worsening bone erosion and joint damage in RA.

As a chronic inflammatory disease, RA involves the interaction of multiple cell types that communicate through cytokines, chemokines, and direct cell-to-cell contact. The inflamed synovial joint in RA is rich in infiltrated cells, including monocytes, B- and T-lymphocytes, neutrophils, etc., and together with the FLSs, they produce a large amount of extracellular vehicles (EVs). Recently, the importance of EVs in the RA pathology was shown. These membrane-bound structures carry proteins, lipids, nucleic acids, and various forms of non-coding RNA (ncRNA), such as micro-RNA (miRNA), long non-coding RNA (lncRNA), and circular RNA (circRNA), all of which are involved in the regulation of gene expression. Elevated levels of EVs in RA patients compared to the healthy donors were observed [151]. A number of studies have shown that EVs could be used as diagnostic markers for RA severity. For instance, the expression of RF on EVs could be associated with significantly higher disease activity scores, like DAS28, ESR, CRP, and VAS, in comparison with RA patients without RF-positive EVs [152]. Moreover, it has been found that EVs from the synovium of RA patients contain citrullinated and carbamylated proteins on their surface, which serve as autoantigens, and their presence is positively correlated with disease activity [153]. These post-translationally modified proteins are critical in breaking immune tolerance and promoting autoimmunity in RA. It was demonstrated that EVs with PTM proteins on their surface can accumulate around blood vessels and form immune complexes that could contribute to atherosclerosis and cardiovascular problems in RA patients [154]. Recently, Buttari et al. demonstrated that EVs isolated from RA patients could promote in vitro DC activation by inducing MAPK and NF-κB activation, leading to the pro-inflammatory DC phenotype expressed by the production of IL-12, IL-1β, and IL-10 [155]. Bone resorption could also be controlled according to the presence of exosomes with high levels of RANKL expression [156].

In addition, on the EVs’ surface are found to be expressed inflammatory molecules like TNF-α, which could promote the secretion of MMP-1 from synovial fibroblasts [157]. EVs also carry different micro-RNAs such as miR-155-5p, miR-1307-3p, miR-323a-5p, and miR-146a-5p, that are associated with the pathogenesis of RA [158]. These miRNAs are involved in the dysregulation of both innate and adaptive immunity, contributing to synovial hyperplasia, chronic inflammation, and joint destruction. These findings highlight the various roles of EVs as key mediators of intracellular communication in RA, participating in the spreading of inflammation, autoantigen presentation, and tissue damage.

However, the initiation of RA involves an interplay among components of the innate and adaptive immune responses, where immune cells (T cells, B cells, mast cells, and dendritic cells) and synoviocytes (macrophages and fibroblasts) are the key players [8]. In general, the RA pathophysiology goes through three distinct stages: (1) the occurrence of immune abnormalities; (2) synovial inflammation and extensive proliferation; and (3) joint deterioration and bone erosion [8]. The mechanisms involved in the initiation and progression of RA are presented in Figure 2.

The initiation of RA has a multifactorial origin including many genetic and environmental factors. Collectively, all of these factors initiate the early development of RA, including post-translational modifications of a wide range of cellular (collagen) and nuclear (histones) proteins, such as the conversion of the amino acid arginine to citrulline (a process called citrullination) through peptidyl arginine deiminase, a type IV enzyme (PADI4; EC: 3.5.3.15) [1,159]. After citrullination or other post-translational modifications (acetylation or carbamylation), the altered modified self-proteins are recognized by the immune system through binding to major histocompatibility complex (MHC) protein heterodimers, especially those containing the shared epitope and that produce antigen-presenting cells (APCs), such as dendritic cells and macrophages, which carry antigens to the lymph nodes [4,25,160]. These APCs activate CD4 T-helper (Th) cells, causing the stimulation and proliferation of B cells, which in turn distinguish into plasma cells and produce autoantibodies such as ACPA (targeting citrullinated proteins) and RF (targeting IgGs). The presence of these and other antibodies can be detected up to 10 years before the clinical disease onset [4,25,161]. The antibodies bind with their fixed complement to form immune complex and release chemotactic factors such as C5a, and consequently, due to chemotactic gradient, inflammatory cells are subsequently recruited to the rheumatoid joint where they are activated and cause localized inflammation [25,162]. APCs such as plasmacytoid dendritic cells (PDC) and myeloid dendritic cells (MDCs), in response to modified protein, express HLA class II molecules, cytokines (such as IL-12, -15, -18, and -23), and co-stimulatory molecules, leading to the activation of synovial T cells. Dendritic cell-derived TGF-β and interleukins (IL-1β, -6, -21, and -23) support Th17 cells’ (pathogenic cells) differentiation and suppress the differentiation of regulatory T cells (Treg), which shifts T cell homeostasis toward inflammation. Activated Th17 cells release cytokines such as IL-17 and IFN-γ to provoke the expansion of the intimal lining and to recruit macrophages, which release interleukins (IL-1, -6, -12, -15, -18, and-23), TNF-α, TGF-β, granulocyte–macrophage colony-stimulating factor (GM-CSF), macrophage migration inhibitory factor (MIF), MMPs, disintegrin, ADAMTs, prostaglandins, leukotrienes, and reactive nitric oxide. Together, they all cause FLS proliferation known as synovial hyperplasia [1,25,33]. Treg cells produce anti-inflammatory cytokines IL-10 and TGF-β and inhibit autoimmunity; therefore, the balance between Th17 and Treg is important in the pathology of RA [163]. The increase in the mass of synovial cells and immune cells in joints leads to pannus formation (thick, swollen synovial membrane with granulating tissue consisting of myofibroblast, fibroblast, and inflammatory cells) [25,164].

This hyperplastic pannus tissue contributes to cartilage damage but may also be responsible for the propagation and systemic spreading of inflammation by migrating between joints or other organs [25,29]. Synovial immune cell infiltration transforms the paucicellular synovium into chronically inflamed tissue [1]. Due to the resulting local hypoxia, new vessels are formed, which facilitate the inflammatory process by increasing the amount of adaptive immune cells, especially CD4+ Th cells infiltrating the synovial sublining. Lymphocytes infiltrate, accumulate, and form aggregates, which may lead to the development of germinal centers that facilitate local T cell–B cell interactions. In these ectopic germinal structures are specific pathologic follicular helper T cells (Tfh), which promote B cell responses and (auto)antibody production within pathologically inflamed non-lymphoid tissues [1,165]. It has been established that specific pathogenic infiltrating immune cell subsets, such as IL-1β-positive pro-inflammatory monocytes, autoimmune-associated B cells, and peripheral helper T (Tph) cells sharing similarities with Tfh cells, distinct subsets of CD8+ T cells, as well as mast cells, contribute to the inflammatory pattern of the RA synovial lining/sublining [166,167,168,169,170,171]. Finally, the invasive and destructive FLS-front of synovial tissue (called the pannus) attaches to the articular surface and contributes to local matrix destruction and cartilage degradation. The chondrocytes of the damaged articular cartilage contribute to the vicious cycle of cartilage degeneration by inducing inflammatory cytokines, such as IL-1β and TNF-α, as well as MMPs and nitric oxide (NO). Additionally, FLSs negatively affect the subchondral bone through the activation and maturation of bone-resorbing osteoclasts. Osteoclasts are highly responsive to autoantibodies; pro-inflammatory cytokines, in particular TNF, IL-1, and IL-6; and more importantly, the receptor activator of nuclear factor kappa B ligand (RANKL), which is the key regulator of osteoclastogenesis. RANKL binds to its receptor, the receptor activator of nuclear factor-B (RANK), and activates osteoclasts, leading to an enhancement in bone resorption and destruction. Conversely, osteoblasts that play a key role in the regulation of anabolic bone metabolism produce bone matrix constituents, induce bone matrix mineralization, and modulate osteoclasts through the production of osteoprotegerin (OPG) [32]. Although osteoblasts produce OPG, which is a decoy receptor for RANKL and results in protection from bone destruction by osteoclasts, they also generate RANKL and M-CSF, both of which contribute to osteoclastogenesis. Imbalanced bone remodeling both in the subchondral and periarticular bone of joints leads to bone erosions and periarticular osteopenia; generalized bone loss is a general feature of established RA [1,172].

## 2. Effector Cells Involved in Rheumatoid Arthritis Pathology

Rheumatoid arthritis has a complex pathogenesis, and the activation of cells of both innate and adaptive immunity and aberrant cytokine production are the main events responsible for the generation and development of pathological immune response. Consequently, synovial inflammation leading to massive destruction of cartilage and bone structures is observed. The pathogenesis of RA involves a diverse array of effector cells, including T cells, B cells, macrophages, and fibroblasts, each contributing to the inflammatory milieu and tissue damage observed in affected individuals [173,174].

The association of the particular MHC-II alleles with RA chronic inflammation supports the role of CD4+ T cells. Disease-associated HLA-DR alleles, particularly those with the “shaped epitope”, may present arthritis-related peptides, such as those modified by citrullination, leading to the stimulation and expansion of autoantigen-specific CD4+ T cells in the joints and lymph nodes [175].

As in the regular immune response, the autoimmune process starts with the presentation of the processed antigen on APCs, triggering the activation of naïve T lymphocytes. This activation includes the proliferation and secretion of cytokines such as IL2, IFN-γ, TNF, and IL-4, which leads to the modulation of the immune system through the differentiation of various subsets of cells like T helper (Th)1, Th2, T follicular helper (Tfh), Th17, and regulatory T (Treg) cells. The Th1/Th2 imbalance is thought to exacerbate the inflammatory processes in RA as a lack of Th2-mediated regulation allows for unchecked Th1-driven inflammation. In particular, CD4+ T helper cells and CD8+ cytotoxic T cells play pivotal roles in RA pathology. A notable increase in the ratio of CD4+ to CD8+ T cells has been documented in the blood of RA patients, suggesting a skewed immune response favoring CD4+T cell activation [176]. Furthermore, studies have shown that effector memory CD8+ T cells are significantly altered in RA, with a decrease in their numbers in peripheral blood, potentially due to their migration to inflamed tissues [177]. This migration is indicative for their pathogenic role because they can produce pro-inflammatory cytokines such as IFN-γ, contributing to the inflammatory environment within the synovium [178].

Th17 cells are characterized by their production of IL17 and have emerged as critical players in RA pathology. These cells promote the recruitment of neutrophils and enhance the production of pro-inflammatory cytokines, contributing to joint inflammation and damage [179,180]. Studies have shown that Th17 cells are enriched in the synovial fluid of RA patients, correlating with disease severity and joint destruction [181,182]. The differentiation of Th17 cells is often favored in the inflammatory milieu of RA, which further perpetuates the cycle of inflammation and tissue damage [183]. T follicular helper (Tfh) cells are essential for B cells in the germinal centers, promoting their differentiation into plasma cells that produce high-affinity antibodies. In RA, the frequency of circulating Tfh cells is significantly elevated, and these cells are associated with increased levels of autoantibodies, such as anti-citrullinated protein antibodies [184,185]. The presence of Tfh cells in the synovial tissue of RA patients suggests that they play a direct role in sustaining the autoimmune response and contributing to the chronicity of the disease.

Regulatory T (Treg) cells are typically involved in maintaining tolerance and preventing excessive inflammation. In the RA, their functional capacity is often compromised. This dysfunction leads to the inadequate suppression of effector T cell responses, allowing for the persistence of inflammation and autoimmunity [186].

Macrophages are central to the inflammatory processes in RA, acting as a major producers of pro-inflammatory cytokines such as TNF-α and IL-1β. They are found in increased numbers within the synovial tissue, correlating with disease severity and joint erosion [187,188]. Additionally, macrophages expressing heparin-binding EGF-like growth factor (HB-EGF) have been shown to enhance fibroblast invasiveness, further exacerbating joint destruction [189,190].

B cells contribute significantly to RA through the production of autoantibodies and pro-inflammatory cytokines. The dysregulation of B cell subsets, including the expansion of RANKL+ effector B cells, has been observed in RA patients, indicating their involvement in osteoclastogenesis and joint destruction [191]. The efficacy of B cell depletion therapies, such as Rituximab, reinforces the pathogenic role of B cells in RA as these cells not only produce antibodies but also act as antigen-presenting cells, perpetuating the inflammatory cycle [192]. Furthermore, the presence of FcRL4+ B cells in the synovial fluid suggests a unique subset that may contribute to the inflammatory process through enhanced activation and cytokine production [193].

Fibroblast-like synoviocytes are critical effector cells in RA, contributing to the formation of the invasive pannus that characterizes the disease. These cells are activated by various inflammatory mediators and exhibit increased production of matrix metalloproteinases (MMPs) and cytokines, facilitating tissue remodeling and joint destruction [194,195]. The interaction between FLSs and infiltrating immune cells, particularly macrophages, creates a feedback loop that sustains inflammation and joint damage [189,190]. Recent studies have identified functionally district subsets of FLSs that correlate with disease activity, suggesting that targeting these cells may offer therapeutic potential [194].

### 2.1. Cytokines and the Impact on Effector Cells

Experimental studies on RA patients as well as on different mouse models point to the role of interactions between cytokines and effector cells in disease progression [196,197]. Genetically modified TNF-α transgenic and IL-1Ra knockout mouse models of RA showed the importance of TNF-α and Th17 in the activation of the immune cells [198,199]. RA can also be induced by the administration of different agents generating local or systemic inflammation. In the most frequently used model of RA–collagen-induced arthritis, the presence of collagen type II-specific CD4+ T cells leads to the activation of collagen-specific B lymphocytes together with the secretion of various pro-inflammatory cytokines (TNF-α and IL-6), although the anti-inflammatory IL-10 and IL-1 receptor antagonist can also be detected in the affected tissues [197,200]. In an antigen-induced model of RA, the antigen binds directly to the cartilage and induces inflammation with CD4^+^CCR6^+^ T cells via IL23 secretion [201]. Another mouse model of RA is proteoglycan-induced arthritis in which scheduled injections of human cartilage proteoglycan resulted in the synovial inflammation and infiltration of immune cells. CD4+ T cell immune response was observed with high levels of TNF-α and IL-1β, -6, and -12 in experimental mice [202]. Finally, RA can be induced in various mouse strains by the administration of specific anti-collagen type II monoclonal antibodies. In this model, the major effector cells appeared to be macrophages and neutrophils, although B- and T-lymphocytes also participated [203].

In RA patients, the role of effector immune cells and cytokine secretion has also been studied. A recent study reported that the accumulation of B lymphocytes in synovium correlates positively with the severity of the disease [204]. B cells are the major source of autoantibodies to various antigens as some antibodies can be detected before the onset of the disease, whereas the presence of others indicates a more severe form of RA [205]. In addition, being effective antigen-presenting cells [206] and producers of a number of pro-inflammatory cytokines [168,207], B lymphocytes stimulate autoreactive T-lymphocyte differentiation and proliferation [208]. RA pathogenesis is also caused by activated T cells, and the massive infiltration of CD4+ T lymphocytes can be observed in inflamed synovial tissues [204]. One particular population—peripheral helper PD-1+CD4+ T cells—has been shown to have a central place in B cell co-stimulation and the secretion of pro-inflammatory cytokines [209]. Although part of the innate immune system, other important players in RA pathogenesis are the neutrophils. Abnormal neutrophil activation due to aberrations in gene expression or in metabolic pathways can lead to inappropriate local degranulation with subsequent chronic inflammation [210]. Finally, macrophages in synovium participate in the development of the inflammatory process by the secretion of pro-inflammatory cytokines, chemokines, and metabolites [211]. In addition, Zec et al. recently reported that synovial inflammation is initiated by lining macrophages in the niches and preferentially attract neutrophils in the inflamed tissues [212].

All of these data demonstrate the complexity of RA pathogenesis and propose interactions between immune cells as well as particular cell populations as new targets for forthcoming therapies.

### 2.2. The Role of Metalloproteinases

Matrix metalloproteinases (MMPs) are zinc-dependent endopeptidases belonging to the superfamily of metzincin proteases, and they are involved in extracellular matrix (ECM) degradation and remodeling. MMPs have a critical role in RA since they degrade ECM to destroy the integrity of synovial, cartilage, and bone tissue through the selective cleavage of many non-matrix components present in the extracellular environment, such as cell surface receptor, cytokines, chemokines, cell–cell adhesion molecules, clotting factors, and other proteinases like binding proteins to be involved in inflammation and immunity in RA [213,214]. The key role of MMPs in immune cell development, function, and inflammatory response make them an attractive target therapy in RA disease [215]. Tissue inhibitors of metalloproteinases (TIMPs) are produced by all connective tissue cells and are firmly and irreversibly bound to the active MMPs and pro-MMPs to form a 1:1 non-selective complex to take part in ECM homeostasis [216]. These TIMPs control tissue breakdown by blocking the activities of MMPs, and when imbalance occurs due to the increased production of MMPs or a lack of TIMPs adequate regulation, the pathogenesis of RA is initiated due to dysregulated tissue remodeling (the fine balance between ECM degradation and production) [217]. The ECM is typically composed of the interstitial connective tissue matrix and a basement membrane (BM). The major hydrolases responsible for tissue breakdown are MMP1, 8, and 13 and membrane-type I metalloproteinase (MT1-MMP), and furthermore, MMP9 degrades type II collagen fragments to generate an immunodominant epitope [218]. MMPs 1 and 13 predominate in RA due to their capacity to restrict the collagen degradation rate, while MMP13 is considered to have a dual role in RA because of its capability to degrade aggrecan and proteoglycan, suggesting that it has a dual role in ECM destruction in RA [219,220]. In an RA mouse model, during joint inflammation, the downregulation of collagens (*COL1A1*, *COL3A1*, and *COL5A1*) accompanied by an increase in metalloproteases (*MMP3* and *MMP11*), and the downregulation of enzymes involved in matrix degradation, such as *ADAMTS1* and *ADAMTS2* (a disintegrin and metalloprotease with thrombospondin motifs), were observed [221]. Similarly, in TNFα-treated decellularized chondrocyte matrix, many different collagens (*COL1A1*, *COL1A2*, *COL2A1*, *COL5A1*, *COL5A2*, *COL12A1*, *COL14A1*) but also aggrecan and chondroadherin were strongly downregulated in the damaged matrix, while matrix-degrading enzymes including *MMP2*, -*3*, -*13*, and *19*, as well as *ADAMTS5* and *ADAMTSL4*, were upregulated [222]. MMPs also participate in bone destruction processes, cell migration and invasion, and cytokine and chemokine processing [223]. MMPs’ participation in joint destruction has been related to three mechanisms: (1) adhesion to chondrocytes, leading to the degradation of collagen and cartilage damage; (2) imbalance being provoked in affected joint homeostasis (through the regulation of inflammatory cytokines and chemokines) and the activation of inflammatory signaling pathways that promote osteoclast differentiation and bone resorption; and (3) the promotion of cell migration and invasive angiogenesis [215]. The FLs are involved in both synovial inflammation and bone erosion in RA through the production of various factors such as TNFα, IL-1β, -6, -8, MMP-1, and MMP-13. Stimulated chondrocytes are able to express various proteases, including MMP-1, -2, -3, -7, -8, -9, -10, -13, and -14; ADAM9, 10, and 17; and ADAMTS4. These are all related to cartilage destruction [73,224]. Interleukin-1β induces a variety of MMPs, including MMP-1, -3, -8, -13, -14, and -29, and activates osteoclasts to break down the cartilaginous matrix [214]. Bone destruction can be effectively ameliorated by inhibiting osteoclast differentiation and regulating multiple signaling pathways, including osteoclast differentiation, IL-17, and TNF, as well as MMP-9 and the protein kinase B (AKT) signaling pathway [225]. In collagen-induced arthritis (CIA) in mice, bone destruction was correlated with the significantly elevated levels of *MMP-2* and *MMP-9* in serum, as well as elevated levels of serum TNF-α, IL-6, and IL-1β [226]. In RA mice, the cartilage destruction process was slowed down through the downregulation of the expression of AKT1, VEGFA, IL-1β, IL-6, MMP-9, ICAM1, VCAM1, MMP-3, MMP-13, and TNF-α [227].

The destruction of the BM of synovial cells enables them to migrate through the tissue and permeate the interior of the joint. It has been reported that MMP2 and MT1-MMP are essential for instigating invadopodia generation, cell migration, and invasion [228]. In CIA-induced arthritis, cartilage destruction and bone erosion were reduced by decreased protein levels of TNF-α, IL-1β, IL-6, iNOS, COX2, MMP-1, and MMP-3 in ankle joint tissue [229]. Inhibitors of epithelial–mesenchymal transition (EMT) significantly reduced the excessive proliferation, migration, and invasive behavior of RA-FLs by a reduction in MMP1, -3, and -13 secretion. In addition, the expression of N-Cadherin (an EMT marker) has also been reduced [230]. RA-FLs migration and invasion mechanisms were significantly inhibited through inhibiting the classical TLR4-NF-κB inflammatory pathway and regulating the dynamic balance of MMP-2/TIMP-2, MMP-9/TIMP-1 [231]. The inhibition of the extracellular signal-regulated kinase (ERK) and c-Jun N-terminal kinase (JNK) signaling pathways (ERK/JNK) significantly attenuated the migration and invasion of RA-FLs through leukocyte Ig-like receptor A3 (LILRA3), which resulted in decreased expression of IL-6, IL-8, and MMP3 [232]. Similarly, the decreased expression of MMP-1 and -3 and pro-inflammatory factors (IL-1 and IL-6) and the elevated level of the pro-apoptotic factor (Bax nd cleaved caspase 3) inhibited RA-FLs invasion via the inactivation of the WNT/β-catenin signaling pathway [233].

Matrix metalloprteinase- 3 and -9 levels are high in arthritic tissues and, together with MT1-MMP, are considered a critical step in initiating the degradation of localized BM during synovial proliferation. Therefore, MMP3 and -9 are mostly targeted in patients with RA and considered as non-invasive biomarkers for active RA [223]. Serum MMP-3 was higher in RA patients with high-grade synovitis than patients with low-grade synovitis and significantly correlated with the synovitis score and the activation of the synovial stroma subscore [234]. The combined higher levels of MMP2, -9, and -13 (degraded type I collagen) is a marker of interstitial matrix degradation [235], while MMP2 and -9 activation is a marker of microfibrial degradation [236]. Plant-derived molecules, such as quercetin and luteolin, have the ability to significantly decrease MMP3 and -9 levels [26,237]. Genistein (the predominant soy isoflavone in legumes) supplementation caused a significant decrease in the levels of MMP9 and pro-inflammatory factors TNF-α, IL-1β, and IL-6 in CIA mice [238]. Epimedium herbs (containing icariin, luteolin, quercetin, and kaempferol) were identified, with IL-1β, IL-6, TNF-α, and MMP-9 emerging as key targets for RA [239]. The mechanism of swertiamarin-loaded nanomicelles (a molecule found in *Gentiana macrophylla* Pall) exhibited its anti-rheumatoid activity by blocking the epidermal growth factor receptor/c-Jun N-terminal kinase/matrix metalloproteinase (EGFR/JNK/MMP9) pathway [240]. Recently, MMP7, together with the alpha-chain of fibrinogen, were identified as RA markers, particularly as predictions in response to methotrexate. Low levels of MM7 and the alpha-chain of fibrinogen were associated with improved clinical outcomes after methotrexate treatment [241]. It is interesting to note that resent findings confirmed that the combination of MMP2, -7, -9, -10, and -12 and TIMP-1 are reliable markers to identify RA, while MMP-7 and -10 are a combinatorial signature in systemic sclerosis [242].

### 2.3. The Role of Angiogenesis

An intensive and important process in the early stage of RA is angiogenesis (the development of new blood vessels), which is regulated by many inducers and inhibitors along with the involvement of pro-angiogenic factors like acidic and basic fibroblast growth factors (FGFs), transforming growth factor (TGF)-β, angiopoietin, placental growth factor, and vascular endothelial growth factor (VEGF) [243,244]. A previously unknown mechanism based on the inhibition of the angiogenic functional module of circHIPK3/miR-149-5p/FOXO1/VEGF was demonstrated to have a significant protective effect on RA-FLS and CIA synovium, thus confirming the cross-talk between circular RNAs (circRNAs) and RA [245]. A broad investigation in RA patients revealed an extensive correlation between pro-inflammatory and proangiogenic profiles of disease activity. The most important angiogenic markers were angiopoietin-1, neuropilin-1, Tie-2, endostatin, platelet factor 4 (CXCL4), interleukin-8 (IL-8, CXCL8), vascular cell adhesion molecule 1 (VCAM-1), VEGF, and placenta growth factor (PlGF) [246]. Another mechanism involved in RA angiogenesis suppression is the hypoxia-inducible factor-1α/ vascular endothelial growth factor/angiopoietin 2 (HIF-1α/VEGF/ANG2) axis [247]. Interestingly, the release of heat shock protein 70 kDa (HSP70) can induce angiogenesis. However, the inhibited proteins in the sphingosine Kinases 1/sphingosine-1-phosphate/sphingosine-1-phosphate receptors/Gα protein subunit (SphK1/S1P/S1PRs/Gαi) pathway prevent HSP70 release and therefore produce an anti-angiogenic effect [248].

### 2.4. The Role of Free Radicals

Free radicals, especially reactive oxygen species (ROS), play vital roles in RA pathogenesis and are key modulators of joint inflammation [249]. When ROS are over-generated and/or the antioxidant system is in imbalance, ROS and its related metabolites are accumulated excessively and cause oxidative stress. Hypoxia and oxidative stress might be important drivers of inflammatory processes in arthritic joints. Excessive ROS accumulation might contribute to somatic mitochondrial DNA (mtDNA) alterations, including mtDNA mutations in the synovial tissue and somatic cells of RA patients and increased mitochondrial mutagenesis associated with reduced oxygen tension (pO_2_) in synovial membranes, which induced a pro-inflammatory mitochondrial phenotype [250,251,252]. It has been reported that ROS production increased lipid peroxidation, protein oxidation, and DNA damage in the peripheral blood of RA patients [253,254]. Some markers of DNA damage, such as 8-Hydroxyl-2-deoxyguanosine (8-oxodG), have been found in the synovial fluids and blood of RA patients [255].

## 3. Current Rheumatoid Arthritis Drug Treatment

The treatment for rheumatoid arthritis is focused on controlling inflammation and relieving pain, with the end goal being low disease activity or remission. To achieve this, the European League Against Rheumatism (EULAR) has listed 10 recommendations for the management of RA with disease-modifying antirheumatic drugs (DMARDs). Furthermore, based on those recommendations, an initiative called treat-to-target (T2T) was developed with the idea that RA treatment should not be universal but rather individually tailored to each patient [166,256].

The overarching principle of the T2T approach is the shared decision making between patient and rheumatologist in the next step in the treatment plan based on the efficacy of previous treatments and treatment goals. An individual therapeutic approach is created by following an algorithm adapted from the 2016 update on the EULAR recommendations. The treatment guideline is divided into three phases, with each phase focusing on a different class of DMARDs based on the stage of the disease upon first diagnosis. Failure to achieve the treatment goal within 6 months of starting the therapy results in the patient advancing to the next phase [257].

### 3.1. Conventional Synthetic DMARDs (csDMARDs)

Glucocorticoids (GCs) are the most widely used anti-inflammatory drugs in the field of rheumatology. Intra-articular injections are used to treat acute disease flares, but this type of treatment has been associated with severe adverse reactions. To prevent this, low-dose oral GCs should be used only for short-term pain relief [258]. For long-term control of inflammation, csDMARDs treatment should be started as soon as rheumatoid arthritis is diagnosed. The chemical structures of some of the main csDMARDs are presented in Figure 3.

### 3.2. Methotrexate

Methotrexate (MTX, Trexall) is usually the first choice for RA treatment as it has been proven to be effective as a low-dose monotherapy. Oral administration usually starts at a dose of 15 mg/week but can be increased to up to 25 mg. The mechanism of action of low-dose MTX is thought to be the inhibition of aminoimidazole-4-carboxamide ribonucleotide (AICAR) transformylase (ATIC). AICAR accumulates in the cell, which leads to increased adenosine release (adenosine is a potent immunosuppressant) [259,260]. Common side effects include hematologic abnormalities, gastrointestinal problems, elevated liver enzymes, fatigue, and nausea [261].

### 3.3. Leflunomide

Leflunomide (Arava) is administered orally 50 mg/week. Its effect is lymphocyte-specific immunomodulation. Leflunomide inhibits the mitochondrial enzyme dihydroorotate dehydrogenase, which plays a key role in the de novo synthesis of the pyrimidine ribonucleotides, which in turn prevents the clonal expansion of activated lymphocytes [262]. The most common adverse effects are diarrhea, elevated liver enzymes, rashes, and hypertension [263].

### 3.4. Sulfasalazine

Sulfasalazine (SSZ, Azulfidine) was designed specifically for the treatment of rheumatoid arthritis. It combines an antibiotic, sulphapyridine, with an anti-inflammatory, salicylic acid, linked via an azo bond. The treatment of RA with SSZ is effective, with the main effects being on the gut microbiota and inflammatory cell functions, although the exact mechanism of action is not completely understood. Adverse reactions include gastrointestinal problems, rashes, and nausea [264].

### 3.5. Biologic DMARDs (bDMARDs)

If the patient continues to display moderate disease activity without poor prognostic markers, the therapy should be adjusted to a combination of csDMARDs instead of monotherapy. At the appearance of poor prognostic markers and failure to achieve low disease activity due to csDMARDs’ lack of efficacy and/or toxicity, the therapy should move forward to phase II—using biologic DMARDs. Those agents are highly specific and target pathways of the immune system, so the screening and treatment of any latent infections are highly advised before starting any bDMARDs [213].

### 3.6. Tumor Necrosis Factor-Alpha Inhibitors (TNFis)

TNF-α is a cytokine with both pro-inflammatory and immunoregulatory functions. In the context of rheumatoid arthritis, dysregulated TNF-α was the first cytokine to be proven to directly cause tissue destruction as well as lead to the overproduction of other pro-inflammatory cytokines. Hence, blocking TNF-α signaling is one of the most effective ways to slow down disease progression [265,266].

Since 2000, five TNFis are available. Each antibody has different molecular structure, administration, and dose, but all have the same effect—binding soluble and/or membrane-bound TNF-α. This is summarized in Table 1.

### 3.7. Interleukin-1 Inhibitor

Anakinra (Kineret) is a recombinant human IL-1 receptor antagonist administered subcutaneously at a dose of 100 mg once a day. Studies on the effectiveness of Anakinra suggest that it might be dependent on the cytokine profile of the patient and that the drug works better in combination with other DMARDs [268,269].

### 3.8. Interleukin-6 Receptor Inhibitor

Besides TNF-α, IL-6 is the other cytokine proven to have a key role in the pathogenesis of rheumatoid arthritis. Tocilizumab (TCZ, Actemra) is a humanized recombinant IgG monoclonal antibody that binds to the soluble and membrane-bound IL-6 receptor. It is injected intravenously at a dose of 8 mg/kg every 4 weeks. Studies have shown that Tocilizumab is effective and safe as monotherapy or in combination with other DMARDs with both short-term and long-term effects regardless of the stage of the disease [270,271].

### 3.9. Anti-CD20 Antibody

CD20 is a membrane-bound molecule on the surface of B cells which plays a role in their development and differentiation into plasma cells. Rituximab (RTX, Rituxan) is a chimeric monoclonal antibody originally created to bind to the CD20 molecule on the surface of peripheral B cells in blood cancer patients, but since then, it has been proven effective for treating RA by depleting B cells in the synovium as well. Administration is carried out intravenously with two 1000 mg doses 2 weeks apart [272,273].

### 3.10. Targeted Synthetic DMARDs (tsDMARDs)

Targeted synthetic disease-modifying anti-rheumatic drugs (tsDMARDs) represent a newer class of therapeutics designed to selectively inhibit intracellular signaling pathways, specifically the Janus kinase/signal transducer and activator of transcription (JAK/STAT) pathway. JAK/STAT plays a critical role in both innate and adaptive immunity by mediating cytokine signaling. It is involved in regulating various immune cell functions, including proliferation, differentiation, and survival, as well as inflammatory responses [274].

Compared to bDMARDs, which are large, complex proteins, tsDMARDs are small molecules and do not require injection or infusion, which makes them potentially more convenient. The clinical application of tsDMARDs has expanded due to their oral administration and rapid onset of action, offering convenience and efficacy [275].

Despite their advantages, tsDMARDs carry risks that necessitate careful patient selection and monitoring. Adverse effects such as infections and an increased risk of blood clots have been associated with JAK inhibitors, leading regulatory agencies to issue safety warnings and restrict use in certain populations. As such, guidelines now recommend stratifying patients based on individual risk profiles before initiating tsDMARD therapy.

The tsDMARDs listed in Table 2 are currently in clinical use and include tofacitinib, baricitinib, and upadacitinib. They are used in patients who do not respond adequately to csDMARDs or bDMARDs and are often integrated into treatment guidelines as second- or third-line options, particularly in moderate to severe cases of rheumatoid arthritis and other inflammatory conditions [276,277].

### 3.11. Mesenchymal Stem Cells (MSCs)

Mesenchymal stem cells, also known as multipotent mesenchymal stromal cells, are a heterogeneous group of cells with a plastic-adherent, fibroblast-like morphology. In vitro, they can differentiate into osteocytes, chondrocytes, and adipocytes. According to the criteria established by the International Society for Cellular Therapy (ISCT), MSCs are identified based on the expression of surface markers such as CD73, CD90, CD105, CD44, CD71, and CD106, while they lack markers associated with hematopoietic and endothelial cells, including CD34, CD45, CD11b, CD14, and CD31 [278,279].

MSCs’ main role in physiology is supporting the function and maintenance of hematopoietic stem cells (HSCs) within the bone marrow. From MSCs, the structural elements of the specialized HSC microenvironment are formed; those are referred to as the “niche”, which is essential for regulating HSC behavior, including their self-renewal, differentiation, and survival. Through direct cell-to-cell interactions and the secretion of various cytokines, growth factors, and extracellular matrix components, MSCs modulate the balance between hematopoietic stem cell quiescence and activation, ensuring a steady supply of blood cells while preventing exhaustion of the stem cell pool [280,281].

MSCs’ therapeutic potential in autoimmune diseases and inflammatory disorders is due to their critical role in suppressing T cell proliferation. Several mechanisms for MSC-mediated inhibition of proliferation have been described, primarily secreting immunosuppressive cytokines like TGF-β and IL-10, which induce a state of energy or functional inactivity in T cells. In addition, MSCs express surface molecules such as PD-L1, which bind PD-1 receptors on T cells, promoting T cell apoptosis [282,283]. Lastly, it has been described that MSCs also influence the polarization of T cells, promoting the expansion of regulatory T cells, which further suppress the immune response and maintain immune tolerance [284].

Preclinical studies in animal models of rheumatoid arthritis have demonstrated that MSCs can effectively reduce inflammation, inhibit immune activation, and promote tissue repair in damaged joints. MSCs migrate to inflamed tissues, where they secrete both anti-inflammatory cytokines, such as TGF-β and IL-10, and pro-inflammatory cytokines, such as TNF-α and IL-6. In models with acute joint damage, MSCs have been shown to differentiate into chondrocytes and osteoblasts, restoring the structural integrity to the affected joints, decreasing bone erosion, and reducing cartilage degeneration [285,286].

A key focus of all preclinical studies is to understand the mechanisms behind MSC homing in inflamed tissues and their interaction with the immune system. Research has been performed to identify factors that influence the effectiveness of MSC therapy, including the source of MSCs (e.g., bone marrow, adipose tissue, or umbilical cord), culture conditions, and delivery methods (e.g., intra-articular injection or systemic administration) [287].

## 4. In Vitro Studies Using Plant-Derived Natural Products for the Management of Rheumatoid Arthritis and Signaling Pathways

Multiple transcription factors and signaling pathways are involved in the pathogenesis of RA; the most important and key pathways are MAPK (mitogen-activated protein kinase), NF-κB (nuclear factor kappa B), PI3/AKT (phosphatidylinositol 3 kinase-AKT, also known as PKB), JAK/STAT (Janus-activated kinase signal transduction and activator of transcription), Wnt/β-catenin (Wingless/Integrated), SYK/BTK (spleen tyrosine kinase)/Bruton’s tyrosine kinase), and Notch [19,20]. Major signaling pathways and their possible modulation by plant-derived molecules are presented in Figure 4.

The NF-κB signaling pathway controls many biological processes, mainly inflammation, which is associated with RA. Inflammation has been intensively observed in the early and late stages of RA and is triggered by NF-κB activation in both T cells and antigen-presenting cells directly or indirectly by extracellular and/or intracellular stimuli (e.g., IL-1β, -6, TNF-α, MMPs, etc.). NF-κB signaling may be activated in two different ways: direct activation (canonical and noncanonical pathways) mediated by the inhibitor of kappa B (IkB) kinase (IKK) and NF-kB-inducing kinase (NIK), respectively, and indirect activation, which is interconnected with other cellular pathways, including MAPK, Rho, and phosphoinositide3-kinase (PI3-K) [27]. NF-kB regulates more than 150 genes involved in anti-apoptosis, cell proliferation, and inflammation and plays a key role in regulating the activation, survival, and differentiation of innate and adaptive immune cells. In RA pathogenesis, dysregulated NF-kB signaling contributes to the activation of both immune and non-immune cells through the transcriptional regulation of inflammatory mediators, including TNF-α; IL-1, -2, -6, -8, -9, -12, -18, and -23; GM-CSF; VEGF; RANKL; MCP-1; MIP-2; CXCL1; CXCL10; RANTES; ICAM-1; VCAM-1; MMPs; and COX-2 [27]. On the other hand, inflammatory cytokines also modulate NF-kB through positive feedback, forming a vicious loop, which intensifies RA development [20]. At the same time, excessive NF-κB activation induces apoptosis of abnormal FLS cells in RA, which further accumulate in joint tissues, and debris adheres to cartilage and bone, exacerbating articular cartilage and bone destruction [20]. Along with that, NF-κB dysregulation activates different types of T cells, differentiates Th1 and Th17 cells by inducing IL-12 production, and promotes IL-17 synthesis in Th17 cells, thereby recruiting neutrophils and monocytes to sites of inflammation. Th17 cells contribute to inflammation by regulating the expression of TNF, IL-1b, IL17, IL-21, and IL-22 [27]. The SYK (spleen tyrosine kinase) is a central molecule of B cell receptor signaling, and the level of phosphorylated SYK in peripheral blood B cells of RA patients has been dramatically increased. Strong positive autoantibodies against citrullinated peptides has also been established [288]. Relevant examples of plant-derived molecules demonstrating their effect in in vitro RA models are presented in Table 3.

The 70% ethanol extract of *Periploca forrestii* Schltr. rich in flavonoids revealed notable anti-RA activity. At concentrations ranging from 25 to 500 ng/mL, the extract mitigated cell injury and reduced cell apoptosis and inflammation through the reduced mRNA expression of *COX-2*, *iNOS*, *IL-6*, and *IL-1β* in TNF-α-stimulated L929, HEK293T, and MH7A human RA-FLSs by inhibiting the activation of NF-κB signaling [159]. The sesquiterpene lactone-enriched fraction from *Xanthium mongolicum* Kitag exhibited the strongest anti-RA activity, which dose-dependently (from 1 to 640 μg/mL) decreased the expression of M1-related genes IL-6, -1β, TNF-α, -12b, and iNOS through the suppression of NF-kB signaling [160]. The ethanol extract of *Achyranthes aspera* L. (with doses ranging from 100 to 300 µg/mL) revealed its potential as a natural anti-inflammatory remedy in the treatment of RA by the downregulation of the mRNA expression levels of inflammatory genes, blocking NF-kB promoter activity induced by TNF-α and the inactivation of two upstream signaling molecules, such as Src and Syk kinases [289]. The mechanism of the anti-RA activity of Siweixizangmaoru decoction (with doses ranging from 12.5 to 400 µg/mL) from Tibet was revealed to include the modulation of TNF-α; IL-1β, -6, -4, and -10; MMP-2, -3, and -9; and MMP-13 through the suppression of JAK2/STAT3 and NF-κB signaling [161]. The Chinese herb *Kadsura coccinea* (Lem.) A. C. Smith also revealed its anti-RA activity through the inhibition of NF-κB and STAT1 signaling [162], while *Nyctanthes arbortristis* L from India reduced the production of various inflammatory mediators and factors in vitro, such as NO, ROS, iNOS, COX-2, TNF-α, IL-6, and -1β, by inhibiting the activation of NF-kB [290]. Investigating the effect of *Geranium Wilfordii* Maxim. in MH7A cells established that its anti-RA activity was due to the modulation of the expression of Bax; Bcl-2; IL-6 and -8; MMP-1; MMP-2, -3, and -9; COX-2; and iNOS through the inhibition of the NF-kB and MAPK pathways [291]. The JAK/STAT pathway is another crucial signaling pathway deregulated in RA and governing cell differentiation and proliferation, with special emphasis on inflammation and immune functions [27].

The JAK family has four members, JAK1, JAK2, JAK3, and tyrosine kinase 2 (TYK2), while the STAT family of TFs consists of seven members, namely STAT1, STAT2, STAT3, STAT4, STAT5a, STAT5b, and STAT6. Upon receptor ligation, JAKs are autophosphorylated and recruit and phosphorylate members of the STAT family and many of the pro-inflammatory cytokines that are highly expressed in RA and are known to be regulated by the JAK/STAT signaling pathway, such as TNF-α; IL-1β, -6, -7, -8, -12, -15, -17, -23, and -32; IFN-γ; and GM-CSF [20,27].

*Tinospora cordifolia* (Thunb.) Miers extract, rich in polyphenols, effectively downregulated the level of pro-inflammatory mediators (IL-6, TNF-α, PGE2, COX-2, and iNOs) and angiogenic factor VGEF by targeting the upstream kinases of the JAK/STAT pathway [292]. The acidified methanol extract of *Pennisetum glaucum* (L.) R.Br., rich in polyphenols, demonstrated strong anti-RA activity in vitro by inhibiting MMP-9 and PTGS2 via the suppression of the JAK2 signaling pathway [293]. The water extract of *Pueraria montana* (Lour.) Merr. suppressed inflammation, migration, and invasion in the human rheumatoid fibroblast-like synoviocyte line MH7A by modulating the expression of Bcl-2; Cas-3; Cas-9; MMP-1 and -9; IL-6 and -8; -1β; TNF-α; and SOCS1 [165]. In single osteoclasts cell culture, the classic Chinese herbal compound Yi Shen Juan Bi Pill relieved the symptoms of RA by regulating the bone immune microenvironment via JAK2/STAT3. In vitro, this compound decreased the number of TRAP^+^ cells and the areas of bone resorption and inhibited the expression of RANK, NFATc1, c-fos, JAK2, and STAT3 while promoting the expression of IL-10 [294]. This was similarly observed in another traditional Tibetan medicine (Shi-Wei-Ru-Xiang pills). In a co-culture system of IL-1β-stimulated synoviocytes or chondrocytes, the inhibition of chondrocytes apoptosis was observed, associated with the attenuation of inflammation by the inhibition of the phosphorylation of p38, Erk1/2, and STAT3 [172]. Jolkinolide B (an ent-abietane-type diterpenoid found in *Euphorbia* plants) revealed its anti-RA potential at a 1 µM concentration through the suppression of TNF-α and IL-6 by decreasing the protein expression level of the JAK2/STAT3 pathway in vitro [295].

**Table 3 ijms-26-06813-t003:** Plant-derived molecules targeting the major cytokines, TFs, and signaling pathways in RA directly or indirectly in vitro.

Molecule	Dose, µM	Cell Line	Targets	Main Findings	Modulated Pathway	Reference
Curcumin	50	MH7A	TNF-α, IL-6, IL-17	Inhibition of migration, invasion, and inflammation	PI3K/AKT	[296]
Emodin	15	L929	IL-6, IL-1β, COX-2	Inhibition of inflammation	NF-κB	[297]
Ginsenoside compound K	30	Isolated FLS	FLUT1, HK2, PKM1, PKM2	Inhibition of glycolysis	NF-κB	[298]
Glytabastan B	3 and 6	SW982	TNF-α, IL-6, IL-8, COX-2, MMP-1	Inhibition of inflammation and invasion	MAPK, PI3K/AKT, NF-κB	[299]
Isobavachalcone	20	MH7A	TNF-α, MAPK13, EGFR, PTGS2, MMP-3	Inhibition of migration, invasion, and inflammation	PI3K/AKT, JAK/STAT	[300]
Kaempferol	10	HFLS-RA	IL-1β, MMP-2 and -9, N-cadherin, vimentin	Inhibition of inflammation and abnormal proliferation	MAPK	[301]
Leocarpinolide B	20	SW982	IL-6, IL-8, IL-1β	Inhibition of proliferation, migration, invasion, and inflammation	NF-κB	[302]
Magnoflorine	10	MH7A	iNOS; COX-2; IL-6; IL-8; MMP-1, -2, -3, -9, and -13	Inhibition of proliferation, migration, and invasion	PI3K/AKT, NF-κB, Nrf-2,	[291]
Nimbolide	1	HIG-82	MMP-2, IL-6, iNOS, COX-2	Reduction in inflammation	MAPK, NF-κB, Nrf-2	[303]
Quercetin	1.5	L929, HEK293T, MH7A	COX-2, iNOS, IL-6, IL-1β	Reduction in cell apoptosis and improvement in cell injury	NF-κB	[159]
Sappanone A	40	HFLS-RA	TNF-α, IL-1β, IL-6, IL-10, IL-17A	Inhibition of inflammation	JAK2/STAT3, PI3K/AKT, NF-κB	[304]
Shikonin	1 × 10^−7^	MH7A	VEGF, VEGFR2, TNF-α, IL-1β, PDGF, TGF-β	Inhibition of migration, invasion, and adhesion	MAPK (ERK1/2, JNK, p38)	[305]
Scopoletin	30	HFLS-RA	IL-1β, TNF-α, MMP-3, MMP-9, COX-2, Bcl-2	Inhibition of proliferation, migration, and invasion	NF-κB	[306]
Suberosin	5	RA-FLS	IL-6, IL-1β, TNF-α, IL-8, MMP-1, MMP-3, MMP-9, MMP-13	Inhibition of inflammation	JAK/STAT	[307]
Tectoridin	50	HFLS-RA	IL-1β, IL-6, COX-2, iNOS	Inhibition of inflammation	MAPK (ERK1/2, JNK, p38)	[308]
Umbelliferone	20	HFLS-RA	IL-1β, TNF-α, MMP-3, MMP-9, COX-2, Bcl-2	Inhibition of proliferation, migration, and invasion	NF-κB	[306]
Wilforine	0.4	Isolated FLS	IL-1β, IL-6, TNF-α, CCND1, GSK-3β, c-Myc, MMP-3	Inhibition of inflammation and abnormal proliferation	Wnt11/β-catenin	[309]

The PI3K/AKT pathway regulates proliferation, metabolism, angiogenesis, and cell survival and is correlated with the occurrence and development of RA [20]. This pathway participates in the abnormal proliferation of FLS cells and synovial inflammation by stimulating the expression of inflammatory molecules like IL-1β, -6, -17, -21, and -22 and TNF-α [20]. Abnormal PI3K/AKT pathway activation stimulates the expression of VEGF and HIF-1α to promote angiogenesis, which not only disturbs the nutrition processes in the synovium but also promotes glycolysis [30], and diverse inflammatory mediators are released [20]. PI3K/AKT activates mammalian target of rapamycin (mTOR) and further inhibits autophagy in FLS, promoting the continuous abnormal proliferation of synovial cells, and it is also critical for the survival and differentiation of osteoclasts, aggravating RA [20].

*Hedyotis diffusa* Willd modulated its anti-inflammatory targets (TNF-α, IL-6, IL-17, and IL-10) in vitro through the suppression of the PI3K/AKT signaling pathway, thus revealing its anti-RA capacity at varying concentrations between 0.5 and 2.0 mg/mL [20]. The ethanolic extract of *Ammopiptanthus nanus* (M. Pop.) Cheng f. inhibited the PI3K/AKT/NF-κB pathways closely related to RA by suppressing the expression of inflammatory cytokines IL-1β, COX-2, and iNOS in vitro when used in a concentration between 1 and 30 µg/mL [310].

The MAPK signaling pathway plays a key role in the pathological process of RA in terms of the regulation of various cellular activities, including gene expression, metabolism, migration, survival, cell cycle progression, apoptosis, and differentiation, and its over activation is closely correlated with the articular cartilage destruction and inflammatory hyperplasia of synovial tissues. P38 MAPK, extracellular signal-regulated kinase (ERK), and c-Jun N-terminal kinase (JNK) are the three main subfamilies of the MAPK pathway. ERK1 and ERK2 are important for the regulation of cell differentiation, proliferation, and survival. On the other hand, the main effect of JNK MAPKs in RA is cartilage destruction mediated by MMPs. Similarly, P38 is linked to the inflammatory response in RA and activates many protein kinases and transcription factors that play key roles in the regulation of humoral and cellular autoimmune responses [20].

The dried roots of *Lithospermum erythrorhizon* Sieb containing shikonin as a major compound inhibited the migration, invasion, and inflammation process related to RA in vitro through the inhibition of angiogenic and inflammation mediators, such as VEGF, TNF-α, and IL-1β [305]. The phenolic compound 5-hydroxyconiferaldehyde, isolated from *Campanula takesimana*, revealed its anti-RA potential by inhibiting the inflammatory response in vitro (inhibition of PGE2, iNOS, TNF-α, COX-2, IL-6, and -1β) through the suppression of several signaling pathways, such as MAPK, NF-κB, and Nrf-2 [311]. The saponin-rich fraction in Rhizoma Panacis Majoris exhibited its anti-RA potential through decreasing the expression of autophagy-related indicators (LC3II/LC3I and Beclin-1) and corresponding signaling pathways, such as MAPK and PI3K/AKT [312].

The Wnt (Wingless/Integrated) signaling pathway takes part in a variety of pathological symptoms such as maintenance, differentiation, proliferation, and self-renewal in RA. The Wnt pathway also plays a key role in synovial inflammation and in the regulation of bone metabolism in RA [20]. A traditional Chinese medicine, Er Miao San (a mixture of Atractylodis Rhizoma and Phellodendri Cortex), revealed promising anti-RA activity in vitro by decreasing angiogenesis and the inflammatory microenvironment in MH7A cells by inhibiting Wnt/β-catenin signaling [313]. The total saponins fraction of *Radix clematidis* has pronounced anti-RA activity, inhibiting the excessive proliferation of FLS during in vitro studies. At concentrations ranging from 0.5 to 1562.5 µg/mL, it inhibited c-Myc, cyclin D1, GSK-3β, and SFRP4 markers through Wnt7b/β-catenin [314].

Notch signaling affects numerous processes of normal cell morphogenesis, including cell proliferation, the differentiation of pluripotent progenitors, apoptosis, and the formation of cell boundaries [20]. The Qi-Sai-Er-Sang-Dang-Song decoction (in concentrations ranging from 0.25 to 4%) inhibited RA symptoms with a focus on inflammation by inhibiting IL-6 and -18 and -1β that are regulated by the Notch/NF-κB axis [315]. In the range of 1 to 30 µM, norisoboldine (an alkaloid compound isolated from Radix Linderae) inhibited synovial angiogenesis in vitro by decreasing VEGF expression and the Notch1 signaling pathway [316].

## 5. In Vivo Studies Using Plant-Derived Natural Products for the Management of Rheumatoid Arthritis

Rheumatoid arthritis is a chronic autoimmune disorder characterized by various inflammatory processes causing pain, swelling, redness, and malfunctioning of joints [317]. The etiology of RA represents a complex combination of genetic and epigenetic predispositions and environmental factors [318]. Despite the long list of medications for RA, most of them bring only partial relief to patients; thus, new therapeutic agents for treating the sources of disease and not just the symptoms are still needed [319,320].

The most frequently applied drugs for RA treatment are conventional synthetic disease-modifying anti-rheumatic drugs (csDMARDs), among which methotrexate (MTX) is the recommended first-line therapy for RA; biological DMARDs (bDMARDs) and targeted synthetic DMARDs (tsDMARDs) are also available [321]. Treatment with csDMARDs ideally comprises MTX plus low-dose glucocorticoids. However, the response to MTX varies, and only after 6 months of therapy, 43% of patients are classed as non-responders to MTX [322]. bDMARDs include targeting monoclonal antibodies against TNF-α, IL-6, soluble receptors for TNF, and T cell co-stimulation. tsDMARDs are inhibitors of the Janus tyrosine kinase family (JAK), which targets the intracellular signaling of type I and II cytokines [321]. In addition to poor response, many patients often experience adverse events (AEs) such as gastrointestinal AEs and/or elevated liver enzymes [322], liver fibrosis, myelosuppression, and pneumonitis [323] and have a higher chance of developing infections, tuberculosis, and certain malignancies, such as lung, breast, and skin cancers and lymphoma [324].

Therefore, there is not only a clinical need to identify patients at a high risk of experiencing non-response to DMARDs but also to identify novel effective RA therapeutics without any side effects. In this regard, natural products with anti-inflammatory activity represent promising adjuvant agents or alternatives to RA therapeutics [4].

Experimental therapy with newly developed drugs in humans is limited for technical and ethical reasons. In this case, rodent models are very useful because of their low cost, homogeneity of the genetic background, and ease of handling. Although animal models mimic a part of a disease and never duplicate or reproduce the entire human pathology, they are very useful to test the hypothesis of the global effect of a molecule.

Animal models of RA can be classified into two broad categories: (A) induced animal models for RA and (B) genetic models of RA. Induced models vary according to the chemical agent with arthritogenic properties used, for instance, collagen, pristane, complete Freund’s adjuvant (CFA), and cartilage oligomeric matrix protein (COMP). In the case of genetic models, the animals are either deficient in (knockout) or transgenic for a specific gene of interest [325]. Genetic models have a valuable role in arthritis research. Information regarding the role of a particular depleted or introduced gene provides valuable insights into the process of inflammation and regulation. These models serve as tools to study the effect of therapeutics in mice prone to developing joint inflammation spontaneously and to understand varying manifestations of autoimmunity in general. In the present study, we summarized the widely used rodent models of RA to test the therapeutic potential of the biologically active compounds.

### 5.1. Collagen-Induced Arthritis Model

Known as a gold standard of in vivo models, the collagen-induced arthritis (CIA) model shares many similarities with the pathological and immunological features of human RA. The clinical signs of this polyarthritis model are inflammation of the synovium, cartilage destruction, and bone erosion [326]. Antibodies play an important role in the inflammatory phase of CIA. Immunization with heterologous type II collagen in complete Freund’s adjuvant (CFA) to genetically susceptible mice strains with MHC haplotypes H-2^q^ or H-2^r^ (DBA/1, B10.Q, and B10.RIII) leads to the generation of autoantibodies against self-antigens and collagen [200]. It was also shown that C57BL/6 (B6; H-2^b^) [327] can develop CIA with a high incidence (60–70%) and sustained severity dependently on B and CD4 + T cells. Susceptibility to CIA requires not only MHC class II haplotype restriction but also TCR and non-MHC susceptibility genes. This is also confirmed by the fact that the passive transfer of collagen type II-specific T cells does not induce severe inflammation compared to the transfer of collagen type II-specific antibodies. The dominant pathological role of Th1 and Th17 cells is well known, but the antibodies against collagen type II seem to have a primary role in the immunopathogenesis of this model. CIA is an important rodent model for the analysis of non-MHC genes and their role in RA development [200].

### 5.2. Collagen Antibody-Induced Arthritis Model

The passive transfer of a commercially available monoclonal antibody cocktail to collagen type II presents an alternative to CIA, significantly reducing the study duration and cost while increasing disease induction and synchronicity among individual animals. This model can be induced in MHC haplotypes that are not susceptible to CIA. Terato et al. [328] showed that through i.v. injection of a combination of four different monoclonal antibodies, antibody-mediated CIA can be induced. Further research by Nutty Nandakumar and Rikard Holmdahl [329] has improved the antibody cocktail by selecting epitope-specific antibodies, which are critical for their pathogenicity. The combination of a monoclonal antibody cocktail and LPS led to the development of severe and persistent arthritis to nearly 100% of the mice. The following advantages of the CAIA model can be emphasized: the length of the development (typically within days instead of weeks as in the classic CIA model); synchronization between the animals into the group (the onset of the disease is 2 days after LPS injection); and susceptibility (not only CIA-susceptible mice strains but also some CIA-resistant mice, such as Balb/c and C57BL/6) [330]. The pathology of this model and the induction of inflammation are expressed in the formation of immune complexes, with collagen type II on the cartilage or synovium and the activation of the classical and alternative complement pathways. Additionally, antibody complexes may also activate the monocytes in the joint via Fc receptors, and the release of pro-inflammatory cytokines (i.e., TNF-a and IL-1) recruit neutrophils and macrophages [331]. This model of RA can be used to study inflammatory mechanisms in arthritis and to screen candidate therapeutic agents.

### 5.3. Adjuvant Induced Arthritis Model

The adjuvant induced arthritis model (AIA) presents an efficient way to enhance both the cell-mediated and humoral immune responses towards the antigens by emulsifying them in complete Freund’s adjuvant (CFA) [332]. AIA is severe, sub-chronic arthritis characterized by persistent inflammation with numerous systemic alterations such as synovial hyperplasia, cartilage and bone damage, and edema and deformation. Massive leucocyte infiltration leads to increased chemokine and cytokine levels (as IL-1 and TNF-α) and the release of reactive oxygen species (ROS) [333]. AIA is suitable as a model for screening and testing anti-arthritic drugs. Its manifestations are very similar to those in human RA [334].

### 5.4. Pristane-Induced Arthritis Model

Pristane-induced arthritis (PIA) is severe and a chronic inflammatory model. The introduction of Pristane, a synthetic mineral oil (2,6,10,14-tetramethylpentadecane), induces acute arthritis, a massive formation of osteoclasts, bone erosion and new bone formation, and inflammatory cell infiltration. It is characterized by prolonged and delayed clinical features (from weeks to months). Serological parameters such as rheumatoid factor (RF), antibodies against heat shock proteins and collagen types I and II, and elevated levels of cartilage oligomeric matrix protein can be analyzed. The development of PIA is joint-specifically regulated by T cells and by MHC genes and fulfills the clinical criteria for RA. This model is useful for the validation of new drug candidates [334,335,336,337].

In recent years, different herbal extracts have been reported to have a beneficial effect on RA development in animal models of the disease. The rhizome of *Acorus gramineus* Sol. ex Aiton (a widely used plant in traditional Chinese medicine) has shown a therapeutic effect in aCIA mouse model of RA—it decreases IL-6 and TNF-α and ameliorates swelling of the hind limb [338]. The fruit peel extract of *Annona squamosa* L contains various immunomodulating chemical substances that have anti-RA functions–decreased leucocyte levels in the serum and decreased necrosis in the paws were observed after its administration in CFA-injected mice [339]. Another report demonstrated the beneficial effect of *Saururus chinensis* (Lour.) Baill leaves extract on type II collagen-induced arthritis in mice [340]. The authors concluded that *S. chinensis* extract administration has a positive effect on different RA manifestations—increased serum levels of IL-6 and TNF-alpha, swelling of the hind limbs, and the accumulation of inflammatory cells in the synovial membrane. In their research, Kim et al. investigated the therapeutic potential in RA of an extract mixture of two folk remedies—*Cudrania tricuspidata* (Carrière) Bureau and *Stewartia koreana* Maxim. The results show that treatment with the extract mixture has a strong anti-inflammatory effect as it reduced nitric oxide levels, tumor necrosis factor, IL-6, and IL-1 levels [341]. Another extract, rich in antioxidants, catechins, and proanthocyanidins, is grape seed proanthocyanidin extract. This extract expressed a therapeutic effect in a CIA mouse model of RA by regulating the TLR4-MyD88-NF-κB signaling pathway, which further resulted in an improvement in clinical features and joint histology [342]. Thorn extract of *Gleditsia sinensis* in combination with *Lactobacillus casei* had anti-inflammatory effects in a mouse model of type II collagen-induced RA [343]. Amelioration was observed in decreases in serum nitrite and total cholesterol as well as in pro-inflammatory cytokine (TNF-α and IL 6) levels. Another interesting observation is the report of Allam et al. concerning the therapeutic effect of the phenolic substance ellagic acid in CFA-injected mice [344]. The authors clearly demonstrated that treatment with ellagic acid attenuated paw swelling and bone dysfunction by modulating pro- and anti-inflammatory cytokines.

In addition, the beneficial effect of different plants has been reported in rat models of RA. Wood bark extract from the fruit durian (*Durio zibethinus* Murr.) has shown a therapeutic effect in CFA-injected rats—including improvements in hind limb swelling, histopathological changes, and the suppression of iNOS expression due to the presence of various phenols, alkaloids, tannins, terpenes, saponins, and flavonoids in the extract [345]. Root and leaf extracts of *Chloranthus serratus* (Thunb.) Roem. & Schult. (a traditional Chinese medicine plant) also showed a beneficial effect in RA treatment. The administration of extract in CFA-injected rats resulted in the inhibition of the secretion of pro-inflammatory cytokines [346]. Finally, ethyl acetate extract from the root of *Caragana sinica* (Buc’hoz) Rehder (herbal product in traditional Chinese medicine) has been reported to have therapeutic efficacy in RA treatment in adjuvant-induced arthritis in rats. The administration of extract resulted in reduced paw swelling and protected bone structures due to the negative regulation of the NF-κB pathway [347].

All of these results point to herbal extracts as promising therapeutic agents in experimental mouse and rat models of RA with unexploited potential for further clinical implications (Table 4).

## 6. Human Clinical Trials Involving Phytochemicals for RA Treatment

Polyphenols represent a diverse class of plant-derived compounds that have demonstrated anti-inflammatory and antioxidant activity. Phytochemicals such as curcumin, tea polyphenols, saffron, ginger, hesperidin, etc., have shown modest clinical benefits, particularly in reducing inflammation markers and improving joint outcomes in patients with RA. They have shown generally good safety. These bioactive polyphenol components show numerous health potential benefits like anti-inflammatory effects, antioxidant activity, anti-cancer, neuroprotective, and wound-healing properties [348]. Due to these benefits, these compounds have been included in a number of clinical trials exploring their therapeutic potential in RA.

In a randomized, double-blinded and placebo-controlled study, it was shown that RA patients (12 patients per group) that received two doses (250 or 500 mg) of a highly bioavailable form of curcumin twice a day for 90 days experienced significantly improved clinical symptoms like C-reactive protein (CRP), the erythrocyte sedimentation rate (ESR), rheumatoid factor (RF) values, etc. In another randomized study involving 45 patients with RA that received curcumin (500 mg) alone, diclofenac sodium (50 mg) alone, and a combination of the two for two months, the patients in all therapeutic groups declared a significant improvement in disease manifestations at the end of the trial, especially in the group treated with curcumin alone. The combination of curcumin and diclofenac has shown slightly less efficacy than curcumin alone. The beneficiality of the curcumin was improved, with a lack of adverse events compared to the diclofenac-treated group.

Beyond its anti-inflammatory effects, some clinical studies have shown the metabolic benefits of curcumin supplementation, such as reduced body weight, the levels of fasting blood sugar, low-density lipoprotein cholesterol, etc. Pourhabibi-Zarandi et al. investigated the effects of curcumin on the metabolic parameters in patients with RA. In a randomized, double-blinded and placebo-controlled clinical trial, 48 women with RA received 500 mg curcumin daily for 8 weeks. After the end of the study the collected data showed significant reductions in body weight, body mass index, and serum levels of high-sensitivity CRP and triglyceride levels compared to the placebo group [349]. Different formulations of curcumin, like nanomicelles [350], a hydrogenated curcuminoid formulation [351], and curcumin in combination with other supplements [352] have been tested, and all of them demonstrated an improvement in the patients’ clinical symptoms.

Ginger is another very powerful polyphenolic component with well-known anti-emetic, anti-fever, anti-cough, anti-inflammatory, anti-diabetic, anti-hyperlipidemic, and anti-cancer properties. More than 40 antioxidants have been isolated from ginger rhizome. A number of studies have shown its anti-rheumatic activity and its effect on the expression of a number of inflammation-mediated genes in RA patients, including NF-κB, PPAR-γ, FoxP3, T-bet, GATA-3, and RORγt [353]. Additional research has shown a significant reduction in CRP, IL-1β mRNA levels, and TNFα mRNA levels [353].

*Crocus sativus* (Saffron) has also shown beneficial effects on patients with RA manifestations without significant side effects. Saffron supplementation for 12 weeks reduced the number of tender and swollen joints, pain intensity, and disease activity. Furthermore, the inflammatory parameters in the serum, including TNF-α, IFN-ɤ, and malondialdehyde, were also decreased [354,355].

Another compound with therapeutic potential for the treatment of RA is quercetin—a flavonoid found in the fruits and vegetables with numerous biological effects. Although its exact mechanisms of action remain to be investigated, in a study with 50 women with RA that received quercetin (500 mg/day) or placebo for 8 weeks, significant reductions in plasma hs-TNFα levels and in Disease Activity Score 28 (DAS-28), early morning stiffness (EMS), and morning and after-activity pain were observed [356].

Resveratrol, another a naturally occurring polyphenol, has been explored for its potent anti-rheumatic properties. In a randomized controlled clinical trial, RA patients at different stages of disease activity received 1 g of resveratrol daily with conventional treatment for 3 months. The results show that the disease activity score assessment of 28 joints and other serum markers like C-reactive protein, the erythrocyte sedimentation rate, undercarboxylated osteocalcin, MMP-3, TNF-α, and IL-6 were significantly influenced [357].

Numerous other phytochemicals with potential anti-rheumatic properties are under investigation, as summarized in Table 5.

## 7. Conclusions and Future Perspectives

Rheumatoid arthritis is a chronic autoimmune disorder characterized by inflammation of the joints, resulting in pain, joint swelling, and finally joint damage and bone erosion. Profound insights into RA etiology reveals that many genetic, epigenetic changes, post-translational modifications, and environmental factors, including complex interactions at mucosal levels between microbiome and host immune cells, provoke the pathogenesis and progression of RA. Although there are numerous drugs and novel biological therapies available, they have several limitations and drawbacks, including disease recurrence and adverse effects due to long-term use. Rheumatoid arthritis has transitioned from a highly disabling disease for which no effective remedies existed to a disorder that can be controlled well, with many patients reaching remission. However, many unmet needs remain since not all patients reach sustained clinical remission, and even about 25% still suffer from moderate or even high disease activity; some patients still have not reached low disease activity; and great similarity of response rates to the various targeted therapies has been frequently observed. Therefore, it is still necessary to uncover the cause(s) of RA in order to assure cure or at least prevention; novel therapies need to be established; and different yet unidentified signaling pathways in non-responders need to be identified.

Plant-based natural extracts and molecules, including flavonoids, terpenoids, alkaloids, and many others possessing anti-inflammatory, immunomodulatory, antioxidant, and anti-apoptotic activities, have demonstrated an excellent curative effect on autoimmune diseases, including RA, and some of them have already been used in treating RA patients. The major challenges in using medicinal plants are their complexity related to identification of bioactive components, understanding the mechanisms of the phytopharmaceutical activity, and their poor availability. The inclusion of plant-derived extracts and molecules into novel delivery systems, such as nanoparticles, improves their pharmacological and therapeutic properties. Nanoencapsulation seems to be a promising strategy to enhance the permeability and solubility of the molecules, protecting them from degradation and improving their bioavailability. Therefore, this encapsulation allows for precise dosage, prolonged and controlled release of phytochemicals, and reduced dosage regimen and drug toxicity, improving patient compliance. The use of nanocarriers offers an innovative platform for the topical delivery of plant-derived molecules in the treatment of RA. Using different “omics” technologies, such as metabolomics, 2D and 3D in vitro models, and different panels of in vivo models, is also a possibility to overcome some of the challenges associated with using plant-derived products and may finally decrease the failure of potential new therapies in clinical trials. Combining different 3D tissue models with microfluidic devices could be the next-generation in vitro approach to study the complex cross-talk between tissues/organs and the immune system, including the spreading of (auto)immune reactions across different organs. Although conventional RA mice models have broad diversity, they are partially suited to preclinical testing of cell-based therapies. For example, spontaneous mice models are suitable for studying autoimmunity, while induced models are more focused on the effector phases of immune-mediated arthritis. For that reason, translational research in humanized mice models that accurately mirror the autoimmune processes of human RA and permit its modulation by the transfer of human immunoregulatory cells is expected to be a powerful method for the preclinical evaluation of cell-based immunotherapies. Human-based approaches will provide opportunities to identify objective patient-related biomarkers to elucidate disease subtypes and treatment response but will also enable strategies for the management of patients who are “refractory” or resistant to available treatments.

Plant-derived products also contain prebiotic components, whose interaction with the host microbiome might have a significant impact on health and disease. A future task for researchers in the field will be to identify how these parameters interact to trigger an autoimmune inflammatory response, and the development of RA and would further help to optimize the selection of plant-derived products for therapy and define their mechanisms of activity. In the near future, additional research efforts might be invested in combining phytotherapy with stem cell research, clustered regularly interspaced short palindromic repeats, and CRISPR-associated protein 9 (CRISPR-Cas9) genome editing or gene therapy, which would provide a long-term therapeutic advantage for RA patients following the evaluation of safety, ethical, and medical concerns.

Stem cell research offers a promising avenue for treating RA through the reduction in inflammation and the promotion of tissue repair, offering pain relief. MSCs modulate the immune system and support tissue regeneration and are therefore a primary focus due to their ability to differentiate into various cell types. However, clinical trials are limited, and some concerns remain regarding dosage, administration routes, and long-term efficacy. Ethical considerations and the risk of immune rejection also present challenges, particularly with embryonic stem cells. CRISPR/Cas9 technology is being explored to modify stem cells, particularly induced pluripotent stem cells (iPSCs), to produce anti-inflammatory substances or to target specific genes involved in RA’s progression, potentially offering a more targeted and safer approach compared to traditional treatments. Specifically, in RA, CRISPR-Cas9 is being explored to target genes involved in inflammation and bone destruction. While CRISPR-based gene editing offers potential for correcting genetic abnormalities and modulating immune responses in RA, challenges remain in delivery methods, off-target effects, and achieving complex genomic modifications.

## Figures and Tables

**Figure 1 ijms-26-06813-f001:**
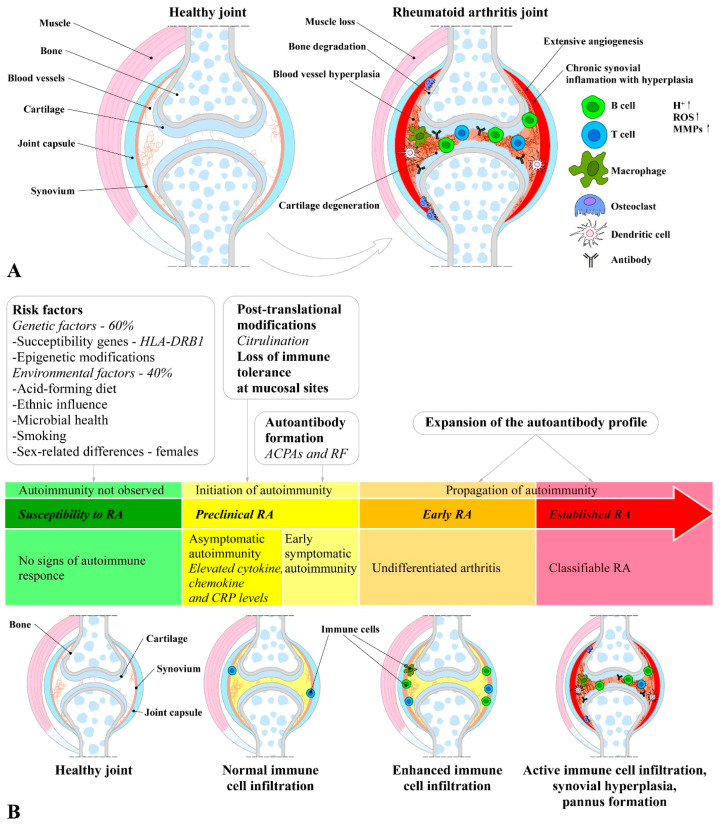
A comparison between healthy and RA joints and RA establishment. (**A**) Due to immune activation, joint swelling is observed, resulting in synovial inflammation, synovial hypertrophy (pannus) formation, osteoclast activation, extensive angiogenesis, bone and cartilage erosion, joint space narrowing, joint structure destruction, and muscle loss. Increased levels of metalloproteinases (MMPs) and reactive oxygen species (ROS) and a decrease in pH are observed as well. The key players are accumulated cells of innate and adaptive immune systems, including T cells, dendritic cells, B cells, macrophages, and osteoclasts. The establishment of RA (**B**). Multiple risk factors (genetic and non-genetic) are required to achieve the threshold for triggering RA. Years before the first instance of subclinical synovitis (inflammation of the synovium) and clinical symptoms, disease progression involves the initiation and propagation of autoimmunity against modified self-proteins. Therefore, once established, RA can be classified according to clinical symptoms.

**Figure 2 ijms-26-06813-f002:**
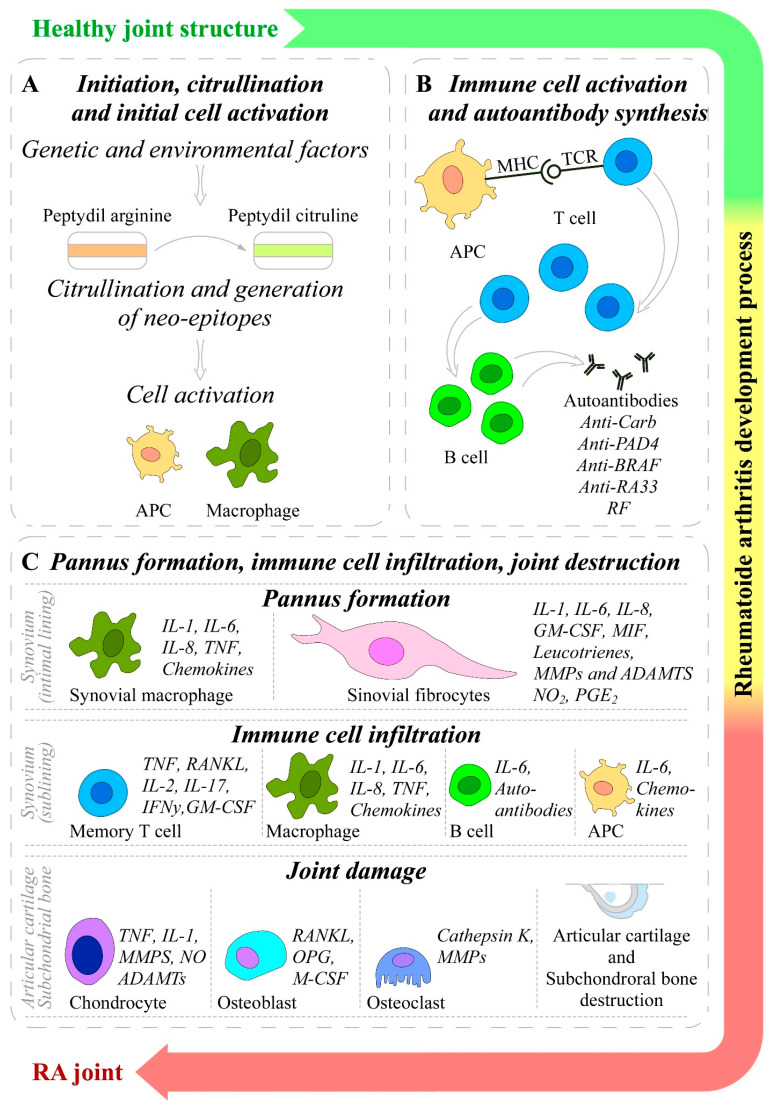
The mechanisms involved in initiation and progression of RA. (**A**) The onset of autoimmunity is supposed to occur in the mucosa (e.g., mouth, lung, and gut) by the production of neo-epitopes resulting from post-translational modifications, such as citrullination or carbamylation. (**B**) The altered peptides (neo-peptides) are recognized by antigen-presenting cells (APCs) of the adaptive immune system and (**C**) are presented to adaptive immune cells in lymphoid tissues, activating an immune response, and they induce autoantibody formation (e.g., ACPA and RF). (**C**) Activated immune cells and complexes activate synovial cells, such as fibroblast-like synoviocytes (FLSs) and macrophage-like synoviocytes of the intimal lining and APCs in the sublining area, thus producing a range of inflammatory factors and resulting in the expansion and formation of the cartilage- and bone-invasive pannus. The activation and infiltration of immune cells (T cells, B cells, and macrophages) of the sublining area further contribute to the excessive production of inflammatory factors, autoantibodies, and synovial vascular leakage, resulting in articular cartilage and subchondral bone destruction due to matrix-degrading enzymes and de-balanced bone homeostasis characterized by an imbalanced RANKL/RANK/OPG system and activated osteoclasts.

**Figure 3 ijms-26-06813-f003:**
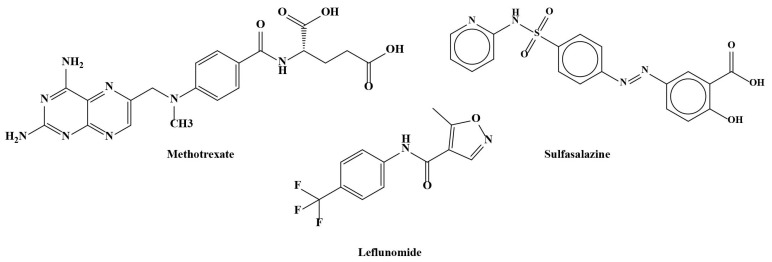
Chemical structures of some csDMARDs.

**Figure 4 ijms-26-06813-f004:**
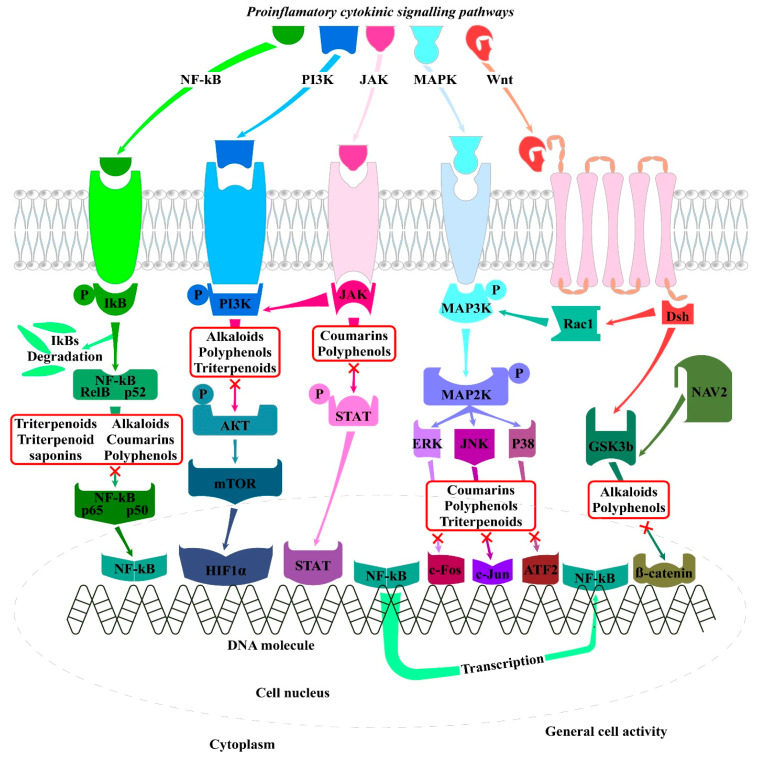
Signaling network in RA and its possible inhibition by plant-derived molecules. Red lines with “x” symbols indicate blocked signaling pathways.

**Table 1 ijms-26-06813-t001:** The molecular structure, administration, and dose of the five FDA approved TNFi antibodies [267].

Antibody	Molecular Structure	Administration and Dose
Infliximab (IFX, Remicade)	Chimeric IgG1 monoclonal antibody	Intravenous injections 3–10 mg/kg every 4–8 weeks
Etanercept (ETN, Enbrel)	Recombinant human fusion protein (TNF-α receptor bound to Fc fragment)	Subcutaneous injections 50 mg/week or 25 mg/twice a week
Adalimumab (ADA, Humira)	Recombinant human IgG monoclonal antibody	Subcutaneous injections 25 mg/twice a week
Golimumab (GOL, Simponi)	Human IgG monoclonal antibody	Subcutaneous injections 50 mg/month
Certolizumab Pegol (CZP, Cimzia)	Recombinant humanized Fab fragment	Subcutaneous injections 400 mg at weeks 0, 2, and 4 followed by 200 mg/every 2 weeks

**Table 2 ijms-26-06813-t002:** The target and dose of the three FDA-approved JAK inhibitors [275].

JAK Inhibitor	Target	Dose
Tofacitinib (Xeljanz)	JAK1 and JAK3	5 mg twice a day or 11 mg once a day
Baricitinib (Olumiant)	JAK1 and JAK2	2–4 mg once a day
Upadacitinib (Rinvoq)	JAK1	once a day

**Table 4 ijms-26-06813-t004:** Modes of action of different herbal extracts reported to have beneficial effect on RA in animal models.

Host	Animal Model	Biological Agent	Mode of Action	Reference
Mouse	Male DBA/1 mice (6 weeks old); CIA (bovine type II collagen + CFA/IFA)	*Acori graminei*	Reduction in inflammation indicators including IL-6 and TNF-α	[338]
Mouse	Male Swiss Albino mice (6 weeks old); AIA (complete Freund’s adjuvant (CFA))	Fruit peel extracts of *Annona squamosa* L.	Decrease in leukocytes in serum	[339]
Mouse	Male DBA/1 mice (6 weeks of age); CIA (bovine type II collagen + CFA/IFA)	*Saururus chinensis*	Reduction in inflammatory cytokines	[340]
Mouse	Male DBA/1J mice (7 weeks of age); CIA (bovine type II collagen + CFA/IFA)	*Cudrania tricuspidata* and *Stewartia koreana*	Decrease in inflammatory cytokine levels, NOS inhibitors	[341]
Mouse	Male DBA/1J mice (6 to 8 weeks of age); CIA (bovine type II collagen + CFA/IFA)	Grape seed proanthocyanidin extract	Inhibition of TLR4/ MyD88/NF-κB signaling pathway	[342]
Mouse	Male DBA/1 mice (6 weeks old); CIA (bovine type II collagen + CFA/IFA)	*Gleditsia sinensis* thorn extract fermented by *Lactobacillus*	Reduction in inflammatory cytokine levels	[343]
Mouse	Male MF1 mice (8 weeks old); AIA (complete Freund’s adjuvant (CFA))	Ellagic acid	Downregulation of pro-inflammatory cytokines and upregulation of anti-inflammatory cytokines	[344]
Rat	Female Wistar rat (3–4 months of age); AIA (complete Freund’s adjuvant (CFA))	Duran wood bark extract	iNOS suppression/NOS inhibitor	[345]
Rat	Male Sprague Dawley (SD) rats (6 weeks of age); AIA (complete Freund′s adjuvant (CFA))	*Chloranthus serratus*	Inhibition of release of inflammatory cytokines and amelioration of antioxidant capacity	[346]
Rat	Male Sprague–Dawley (SD) rats; AIA (complete Freund’s adjuvant (CFA))	*Caragana sinica*	Negative regulation of NF-κB pathway	[347]

**Table 5 ijms-26-06813-t005:** Plant-derived products involved in RA human clinical trials.

Phytochemical	Duration	Dosage	Outcome Measures	Results	Reference
Curcumin	90 days	250 or 500 mg twice a day	ACR response; VAS; CRP; DAS28; ESR; RF	Significant improvement in ESR, CRP, VAS, RF, DAS28, and ACR responses compared to placebo	[358]
	8 weeks	500 mg curcumin or 50 mg diclofenac sodium or their combination twice a day	ACR; DAS28	Curcumin group showed highest percentage of improvement in overall DAS and ACR scores (ACR 20, 50 and 70), and these scores were significantly better than patients in diclofenac sodium group	[359]
	8 weeks	500 mg curcumin daily	Fasting blood samples; anthropometric measurements; dietary intakes; physical activity levels	Significantly decreased HOMA-IR, ESR, CRP, triglycerides, weight, and body mass index compared with placebo	[349]
	12 weeks	40 mg curcumin nanomicelle and placebo capsules 3 times a day	DAS-28; ESR	Within-group values of DAS-28, TJC, and SJC in curcumin nanomicelle and placebo groups reduced significantly compared to baseline	[350]
	3 months	250 or 500 mg hydrogenated curcuminoid formulation or placebo daily	ACR; DAS 28; ESR; CRP; RF	Significant changes in DAS28 (50–64%), VAS (63–72%), ESR (88–89%), CRP (31–45%), and RF (80–84%) values and ACR response for curcumin-treated groups in comparison with placebo	[351]
	6 weeks	0.2% chlorhexidine with scaling and root planing mouthwash; curcumin with scaling and root planing; scaling and root planing alone	CRP; ESR; RF; anti-citrullinated protein antibody; plaque index; pocket depth	Significant reduction in periodontal and RA disease activity parameters was observed from baseline; highest percentage of mean reduction in plaque index and RA parameters from baseline was observed in group treated with curcumin	[352]
Ginger	12 weeks	1500 mg ginger powder or placebo daily	DAS28; gene expression of NF-κB, PPAR-γ, FoxP3, T-bet, GATA-3, RORγt	Statistically significant reduction in DAS28 within ginger group and between two groups; significant increase in FoxP3 gene expression within ginger group and between two groups; T-bet and RORγt gene expression decreased significantly between two groups; in ginger group, PPAR-γ gene expression increased significantly, but difference between two groups was not statistically significant	[353]
	12 weeks	1.5 g ginger per day; placebo	Serum hs-CRP and mRNA levels of IL-1β, IL-2, TNF-α	Ginger powder supplementation caused significant decline in CRP and IL-1β mRNA levels; TNFα mRNA levels reduced in ginger group compared to placebo group but was statistically insignificant; no effects on IL2 gene expression	[353]
Saffron	12 weeks	100 mg/day saffron supplement or placebo	DAS28; ESR; hs-CRP; TNF-α; IFN-γ	Saffron supplementation significantly decreased number of tender and swollen joints; DAS28, hs-CRP, TNF-α, IFN-γ, and malondialdehyde were decreased and total antioxidant capacity was increased	[354,355]
Quercetin	8 weeks	500 mg/day quercetin or placebo	hs-TNFα; ESR; EMS; TSC; SJC; DAS-28; PGA	Significantly reduced EMS, DAS-28, and plasma hs-TNFα levels in quercetin group; no significant differences in TJC and SJC between groups	[356]
Resveratrol	3 months	1 g resveratrol with conventional treatment and control group with regular treatment	Clinical and biochemical markers	DAS-28 was significantly lowered in resveratrol-treated group; ESR, CRP, MMP3, TNF-alpha, and IL6 were also decreased	[357]
Tea polyphenols	8 weeks	2.4 g/day *Stachys schtschegleevii* + 2.4 g/day black tea; placebo—2.4 g/day black tea;	s-CRP; IL-1β; MMPs; DAS28	*Stachys schtschegleevii* intervention caused significant reductions in number and percent changes in tender joints and DAS28 and caused significant MMP-3 reductions	[360]
	6 months	i.v. infusion of Infliximab at dose of 3 mg/kg at baseline, at 2 and 6 weeks later, then every 8 weeks; green tea was supplemented at dose of 4–6 cups/day (60 to 125 mg catechins)	CRP; ESR; DAS28-ESR	Significant decrease (*p* < 0.01) in disease activity parameters (CRP, ESR, and DAS28-ESR) was observed in patients treated with green tea compared with those treated with infliximab or exercise program; TJC and SJC were significantly decreased after 6 months of therapy	[361]
Cinnamon	8 weeks	2000 mg cinnamon powder or placebo daily	Fasting blood sugar (FBS); lipid profile; liver enzymes; CRP; TNF-α; ESR; blood pressure; clinical symptoms	Significant decrease in serum levels of CRP and TNF-α in cinnamon group compared to placebo group; cinnamon intake significantly reduced DAS-28, VAS, TJC, and SJC; no significant changes observed for FBS, lipid profile, liver enzymes, or ESR	[362]
Sesamin	6 weeks	200 mg/day sesamin supplement or placebo	Serum levels of hyaluronidase, aggrecanase, MMP-3; hs-CRP, IL-1β, IL-6, TNF-α, cyclooxygenase-2	Serum levels of hyaluronidase and MMP-3, hs-CRP, TNF-α, and cyclooxygenase-2 decreased significantly in sesamin group compared with placebo group; sesamin supplementation also caused significant reduction in number of tender joints and severity of pain in these patients	[363]
	6 weeks	200 mg/day sesamin supplement or placebo	Serum concentrations of lipid profile; malondialdehyde (MDA); total antioxidant capacity (TAC)	Sesamin supplementation significantly decreased serum levels of MDA and increased TAC and HDL-C levels in patients with RA; means of weight, body mass index, waist-to-hip ratio, body fat, systolic blood pressure, and concentration of other lipid profiles (triglycerides, total cholesterol, and low-density lipoprotein cholesterol [LDL-C]) were also significantly decreased at end of study compared to baseline values	[364]
Olive oil	12 and 24 weeks	3 g/d fish oil omega-3 fatty acids, 3 g/d fish oil omega-3 fatty acids, and 9.6 mL of olive oil; placebo soy oil supplementation	Clinical and laboratory indicators	Significant improvement in joint pain intensity, right and left handgrip strength after 12 and 24 wk, and Ritchie’s articular index for pain joints after 24 wk	[365]
Coenzyme Q10	2 months	100 mg/day CoQ10 or placebo	Serum MMP-1 and MMP-3; DAS-28	Significant reduction was observed in both CoQ10 and placebo groups in medians of serum MMP-1 and swollen joint count and means of DAS-28; significant reductions were only observed in ESR, pain score, and tender joint count in CoQ10 group compared with baseline	[366]
	2 months	100 mg/day capsules of CoQ10 and placebo	MDA; total antioxidant capacity (TAC); IL-6; TNF-α	Serum MDA significantly decreased in supplemented group; suppressed overexpression of TNF-α; no significant difference in TAC and IL-6 levels between groups	[367]
Baicalin	12 weeks	500 mg baicalin or placebo/daily	Lipid profile; cardiotrophin-1 (CT-1); high-sensitivity C-reactive protein (hs-CRP)	Levels of triglycerides, total cholesterol, LDL-cholesterol, and apolipoproteins, as well as CT-1 and hs-CRP, were all significantly improved in baicalin group compared to placebo group	[368]
Extract of *Tripterygium wilfordii* Hook F	20 weeks	180 mg/day or 360 mg/day extract or placebo	Disease activity and treatment response were evaluated according to ACR criteria; morning stiffness and serum titers of RF	80% of high-dose group and 40% in low-dose group experienced disease improvement, fulfilling ACR 20% improvement criteria; no patients in placebo group attained these criteria Significant decreases were also found in number of tender joints, number of swollen joints, and physician’s global assessment in low-dose group	[369]
	24 weeks with 18 additional months for monitoring	60 mg TwHF/day; 7.5 mg MTX/weekly; 60 mg TwHF plus 7.5 mg MTX	ACR criteria, HAQ, ESR or serum CRP level, EULAR; cDAI DAS28	Disease activity in patients from combination and MTX monotherapy groups; significant differences in ACR20, EULAR good response, and DAS remission rate at year 2; all treatment groups had decreases in DAS28 and HAQ scores and increases in SF36 scores	[370]
*Uncaria tomentosa*	52 weeks; 2-phase study	First phase (24 weeks), patients were treated with *Uncaria tomentosa* extract or placebo; second phase (28 weeks), all patients received plant extract	Number of swollen and painful joints and Ritchie Index; VAS; HAQ; ESR, CRP, RF, antinuclear antibodies, complete blood count, hepatic and renal variables	*Uncaria tomentosa* extract resulted in reduction in number of painful joints compared to placebo; there was no change in HAQ	[371]
*Punica granatum*	12 weeks	10 mL/day pomegranate extract	Joint status and serum oxidative status (lipid peroxidation, total thiols group, paraoxonase 1 activity)	Pomegranate extract consumption significantly reduced DAS28; reduction in serum oxidative status; increase in serum high-density lipoprotein-associated paraoxonase 1 (PON1) activity	[372]
	8 weeks	500 mg pomegranate extract per day and 500 mg cellulose for control group	HAQ; DAS- 28; CRP; MMP3; MDA; glutathione peroxidase (GPx); erythrocyte sedimentation rate (ESR)	Pomegranate extract supplementation significantly reduced score of DAS28 and tender joint count, pain intensity, and ESR levels compared to placebo; HAQ score and morning stiffness were also decreased; no differences in MMP3, CRP, and MDA levels between two groups	[373]
Garlic	8 weeks	1000 mg garlic or placebo daily	Serum levels of total antioxidant capacity (TAC) and malondialdehyde (MDA); HAQ	Significant increase in serum levels of TAC in garlic group compared with placebo group; MDA levels were significantly decreased in intervention group compared with control group; HAQ scores decreased in garlic group compared with placebo	[374]
*Stephania tetrandra*	12 weeks	10 g *Stephania tetrandra*	Tender joint count, swollen joint count, patient’s assessment of pain, patient’s global assessment of disease activity, physician’s global assessment of disease activity, HAQ, CRP, ACR score IgM rheumatoid factor (IgMRF) concentration	Proportion of granulocytes and granulocyte count in peripheral blood decreased significantly; lipid peroxide and human granulocyte elastase levels of stored plasma declined significantly	[375]
*Andrographis paniculata*	14 weeks	30 mg of andrographolides three times a day; placebo group	Clinical signs and symptoms of pain and swelling evaluated by VAPS, ACR, EULAR, SF36	Decreased intensity of joint pain in active group vs. placebo group; reduction in rheumatoid factor, IgA, and C4	[376]
*Mangifera indica* L. Extract	180 days	900 mg/day Mangifera indica extract combined with 12.5 mg/week methotrexate and 5–10 mg/day NSAIDs and/or prednisone; control group on usual treatment	Tender and swollen joint counts, ESR, DAS 28, VAS, HAQ; treatment’s efficacy demonstrated with ACR criteria	Only patients in MTX *Mangifera indica* extract group revealed statistically significant improvement in DAS 28 parameters with respect to baseline data but no differences were observed between groups; ACR improvements amounted to 80% only in MTX Mangifera indica extract group at 90 days and decreased NSAID administration	[377]
*Artemisia annua* extract	12, 24 and 48 weeks	30 g/day extract of *Artemisia annua* L.; control group treated with leflflunomide and methotrexate	Pain score, tenderness score, number of painful joints, number of swollen joints, HAQ score for quality of life, RF levels, CCP-Ab, ESR, CRP, VAS score	At 24 and 48 weeks, overall efficacy of extract of Artemisia annua L-treated group was significantly higher than control group; tenderness score, number of painful joints, number of swollen joints, HAQ, CRP, RF, and CCP-Ab were significantly improved compared with control group after 48 weeks of treatment	[378]
Black barberry hydroalcoholic extract	12 weeks	2000 mg black barberry extract or maltodextrin as placebo/daily	Cytokines IL-2, IL-4, IL-10, and IL-17 in blood sample; physical activity; dietary intake; disease activity	Black barberry supplementation reduced severity of RA; IL-17 levels decreased significantly after intervention within black barberry group, while IL-10 had significant increase in this group; no significant effect on IL-2 and IL-4 cytokines	[379]
*Nigella sativa* oil extract	8 weeks	1000 mg Nigella sativa oil/daily or placebo	Serum TNF-α, IL-10, and whole blood levels of oxidative stress parameters	Serum level of IL-10 was increased in Nigella sativa group; significant reduction in serum MDA and NO compared with baseline; no significant differences in TNF-α, SOD, catalase, and TAS values between or within groups	[380]

## Data Availability

Not applicable.

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
