# Peer review of "The Therapeutic Potential of Phytochemicals Unlocks New Avenues in the Management of Rheumatoid Arthritis"

_ijms, 2025, doi:10.3390/ijms26146813_

Round 1
Reviewer 1 Report
Comments and Suggestions for Authors
- Line 38. No therapeutic agent, whether natural or synthetic, can genuinely claim to have "no side effects." It would be better to rephrase to "fewer side effects," "reduced toxicity," or "improved safety profile."
- Line 43. While the abstract mentions "research progress in preclinical RA in vitro and in vivo models," it lacks a comprehensive discussion of human clinical trials involving phytochemicals for RA treatment. Including relevant descriptions is recommended.
- Line 79. While the statistical data in Section 1.1 is informative, it only extends to 2020. Incorporating the latest epidemiological data up to 2025 would significantly improve the relevance and timeliness of this section.
- Line 372. It is advisable to update information on air pollution to include recent findings regarding specific components of particulate matter, their sources, and their associations with the risk of RA.
- Line 615. Section 2.2 outlines the critical role of matrix metalloproteinases (MMPs) in RA. It is recommended that the latest research findings on MMP interactions, including any discoveries up to 2025, be incorporated. Additionally, their potential as emerging biomarkers should also be addressed.
- Line 736. The discussion regarding csDMARDs, which include glucocorticoids (GCs), methotrexate (MTX), leflunomide, and sulfasalazine (SSZ), is generally accurate. It would be beneficial to include the chemical structures of these drugs to enhance the presentation of this class of medications.
- Line 766. This section discusses bDMARDs but does not include information on the latest and currently approved targeted synthetic disease-modifying antirheumatic drugs (tsDMARDs), particularly Janus kinase (JAK) inhibitors. These inhibitors are becoming increasingly important in treating RA. To provide a more comprehensive overview of current RA treatment strategies, it is recommended to include an in-depth discussion of tsDMARDs.
- Line 1158. Table 3 is valuable. To enhance its utility, consider specifying the exact animal model used for each extract, as this information is important for understanding the "mode of action."
- Line 1178. In the "Future Perspectives" section, it would be helpful to not only describe how advanced technologies might be used to tackle challenges related to phytochemicals in RA but also to more specifically discuss their limitations and potential breakthroughs, rather than making general statements.
Author Response
Dear Editor-in-Chief,
We would like to express our sincere thanks to yours’ and the reviewers’ constructive and helpful comments. The manuscript has been revised accordingly and our detailed responses to the comments are hence listed below. All changes made in the manuscript are highlighted with track changes. We hope that the revised version is now suitable for reproduction in International Journal of Molecular Science.
Reviewer #1.
- Line 38. No therapeutic agent, whether natural or synthetic, can genuinely claim to have "no side effects." It would be better to rephrase to "fewer side effects," "reduced toxicity," or "improved safety profile."
Answer: Thank you for your suggestion. Relevant rephrases were done in the track changed version, Line 38 and Line 152.
- Line 43. While the abstract mentions "research progress in preclinical RA in vitro and in vivo models," it lacks a comprehensive discussion of human clinical trials involving phytochemicals for RA treatment. Including relevant descriptions is recommended.
Answer: Thank you for your constructive recommendation. We have included relevant discussion on human clinical trials, including a table with examples as an additional section. Lines 1391-1489.
- Line 79. While the statistical data in Section 1.1 is informative, it only extends to 2020. Incorporating the latest epidemiological data up to 2025 would significantly improve the relevance and timeliness of this section.
Answer: Thank you for your constructive comment. The statistics have been updated, Lines 86-94.
- Line 372. It is advisable to update information on air pollution to include recent findings regarding specific components of particulate matter, their sources, and their associations with the risk of RA.
Answer: Thank you for improving the quality of our manuscript. The resent findings regarding air pollution, particulate matter, their sources, and their associations with the risk of RA have been added in Lines 395-442.
- Line 615. Section 2.2 outlines the critical role of matrix metalloproteinases (MMPs) in RA. It is recommended that the latest research findings on MMP interactions, including any discoveries up to 2025, be incorporated. Additionally, their potential as emerging biomarkers should also be addressed.
Answer: Thank you for the recommendation. Recent discoveries on MMPs and their potential as biomarker have been discussed in Lines 857-880.
- Line 736. The discussion regarding csDMARDs, which include glucocorticoids (GCs), methotrexate (MTX), leflunomide, and sulfasalazine (SSZ), is generally accurate. It would be beneficial to include the chemical structures of these drugs to enhance the presentation of this class of medications.
Answer: Thank you for your comment. A new figure 3 has been added.
- Line 766. This section discusses bDMARDs but does not include information on the latest and currently approved targeted synthetic disease-modifying antirheumatic drugs (tsDMARDs), particularly Janus kinase (JAK) inhibitors. These inhibitors are becoming increasingly important in treating RA. To provide a more comprehensive overview of current RA treatment strategies, it is recommended to include an in-depth discussion of tsDMARDs.
Answer: Thank you for your suggestion. A new subsection 3.3 Targeted synthetic DMARDs (tsDMARDs) has been added from Line 1014 to Line 1038.
- Line 1158. Table 3 is valuable. To enhance its utility, consider specifying the exact animal model used for each extract, as this information is important for understanding the "mode of action."
Answer: Thank you for your suggestion. A relevant modification in Table 3, now Table 4 has been done.
- Line 1178. In the "Future Perspectives" section, it would be helpful to not only describe how advanced technologies might be used to tackle challenges related to phytochemicals in RA but also to more specifically discuss their limitations and potential breakthroughs, rather than making general statements.
Answer: Thank you for your constructive comment. Limitations of stem cell research and CRISP/Cas9 has been added at Lines 1553-1567.

Reviewer 2 Report
Comments and Suggestions for Authors
This manuscript of Kalina A. Nikolova-Ganev et al. offers a thorough and well-supported overview of the pathophysiology of rheumatoid arthritis (RA) , along with the potential role of phytochemicals and plant-based compounds as complementary or alternative treatment approaches.
While the manuscript is overall comprehensive, I have a few suggestions that may help enhance its clarity and impact.
-Could you add some limitations using these compounds? such as toxicity, limitations of current phytochemical studies....
-The authors are encouraged to discuss more about the emerging role of autophagy in the pathogenesis of rheumatoid arthritis (RA), particularly in relation to immune cell survival, synovial inflammation, and bone homeostasis and contribution to post-translational modifications (PTMs)—such as citrullination, carbamylation, and acetylation—which are central to autoantigen formation in RA. The role of extracellular vesicles, especially microvesicles, in mediating intercellular communication and perpetuating inflammation in the RA joint microenvironment also deserves mention. Notably, several phytochemicals discussed in the review have been shown to modulate these processes, including autophagy pathways and PTM-related signaling, suggesting their potential relevance beyond classical cytokine suppression.
-The review offers a detailed overview of genetic and hormonal risk factors for rheumatoid arthritis (RA), including its notably higher prevalence in women. In light of this, would the authors consider expanding on the role of sex chromosome dosage in RA susceptibility? Although often under-discussed, conditions such as Klinefelter syndrome (47,XXY) have been associated with increased autoantibody prevalence and a higher risk of autoimmune diseases, including RA. Including a brief reference to this dimension—supported by recent findings such as the 2020 Clinical and Experimental Immunology study—could strengthen the section on sex-related genetic vulnerability and offer further context for sex-biased disease mechanisms.
Author Response
Dear Editor-in-Chief,
We would like to express our sincere thanks to yours’ and the reviewers’ constructive and helpful comments. The manuscript has been revised accordingly and our detailed responses to the comments are hence listed below. All changes made in the manuscript are highlighted with track changes. We hope that the revised version is now suitable for reproduction in International Journal of Molecular Science.
Reviewer #2.
This manuscript of Kalina A. Nikolova-Ganev et al. offers a thorough and well-supported overview of the pathophysiology of rheumatoid arthritis (RA) , along with the potential role of phytochemicals and plant-based compounds as complementary or alternative treatment approaches.
While the manuscript is overall comprehensive, I have a few suggestions that may help enhance its clarity and impact.
-Could you add some limitations using these compounds? such as toxicity, limitations of current phytochemical studies....
Answer: Thank you for the constructive suggestion. Short discussion on advantages and limitations of phytotherapy in RA has been done in Lines 155-160 in the track changed version.
-The review offers a detailed overview of genetic and hormonal risk factors for rheumatoid arthritis (RA), including its notably higher prevalence in women. In light of this, would the authors consider expanding on the role of sex chromosome dosage in RA susceptibility? Although often under-discussed, conditions such as Klinefelter syndrome (47,XXY) have been associated with increased autoantibody prevalence and a higher risk of autoimmune diseases, including RA. Including a brief reference to this dimension—supported by recent findings such as the 2020 Clinical and Experimental Immunology study—could strengthen the section on sex-related genetic vulnerability and offer further context for sex-biased disease mechanisms.
Answer: Thank you improving our manuscript. A relevant discussion has been added from Line 475-511.

Round 2
Reviewer 2 Report
Comments and Suggestions for Authors
The manuscript is well-developed and complete; however, it would be valuable to include insights into the role of autophagy in the generation of post-translational modifications (PTMs) and extracellular microvesicles and how these modifications contribute to disease pathogenesis.
Author Response
Dear Editor-in-Chief,
We would like to express our sincere thanks to yours’ and the reviewers’ constructive and helpful comments. The manuscript has been revised accordingly and our detailed responses to the comments are hence listed below. All changes made in the manuscript are highlighted with track changes. We hope that the revised version is now suitable for reproduction in International Journal of Molecular Science.
Reviewer #2.
The manuscript is well-developed and complete; however, it would be valuable to include insights into the role of autophagy in the generation of post-translational modifications (PTMs) and extracellular microvesicles and how these modifications contribute to disease pathogenesis.
Answer: Thank you for the valuable advice. We have included a relevant information at Lines 537-554, Lines 575-578 and Lines 590-600 (in the track changed version).

Round 3
Reviewer 2 Report
Comments and Suggestions for Authors
The authors have addressed my concerns